# Neurobiological Highlights of Cognitive Impairment in Psychiatric Disorders

**DOI:** 10.3390/ijms23031217

**Published:** 2022-01-22

**Authors:** Anna Morozova, Yana Zorkina, Olga Abramova, Olga Pavlova, Konstantin Pavlov, Kristina Soloveva, Maria Volkova, Polina Alekseeva, Alisa Andryshchenko, Georgiy Kostyuk, Olga Gurina, Vladimir Chekhonin

**Affiliations:** 1Mental-Health Clinic No. 1 Named after N.A. Alekseev, 117152 Moscow, Russia; hakurate77@gmail.com (A.M.); abramova1128@gmail.com (O.A.); soloveva.kr@yandex.ru (K.S.); vmu10121995@gmail.com (M.V.); p249703a@yandex.ru (P.A.); alissia.va@gmail.com (A.A.); kgr@yandex.ru (G.K.); 2Department of Basic and Applied Neurobiology, V. Serbsky Federal Medical Research Centre of Psychiatry and Narcology, 119034 Moscow, Russia; uovnew@mail.ru (O.P.); pkanew@mail.ru (K.P.); olga672@yandex.ru (O.G.); chekhoninnew@yandex.ru (V.C.); 3Department of Medical Nanobiotechnology, Pirogov Russian National Research Medical University, 117997 Moscow, Russia

**Keywords:** cognitions, psychiatric disorder, depression, MCI, schizophrenia, Alzheimer’s disease, biomarkers, SNP, neuroinflammatory, epigenetic regulation

## Abstract

This review is focused on several psychiatric disorders in which cognitive impairment is a major component of the disease, influencing life quality. There are plenty of data proving that cognitive impairment accompanies and even underlies some psychiatric disorders. In addition, sources provide information on the biological background of cognitive problems associated with mental illness. This scientific review aims to summarize the current knowledge about neurobiological mechanisms of cognitive impairment in people with schizophrenia, depression, mild cognitive impairment and dementia (including Alzheimer’s disease).The review provides data about the prevalence of cognitive impairment in people with mental illness and associated biological markers.

## 1. Introduction

Cognitive functions are essential for the normal functioning of every individual, and if they are damaged, daily life is extremely difficult and unproductive. Cognitive functioning refers to multiple mental abilities, including attention, memory, information processing and remembering, problem-solving, reasoning, and decision making. Throughout a person’s life, cognitive performance changes in response to environmental changes. As children become teenagers, their cognition develops, and eventually, cognitive decline occurs as a part of aging. Because central nervous system (CNS) regeneration processes are imperfect, some cognitive functions may become impaired as a part of normal aging due to neuronal degeneration and death. Cognition is the process of receiving and storing information for further use in behavior management. It is the ability to perceive and react, to process information, and to make sense of it. It also involves making good decisions and generating effective behavioral patterns. In other words, better cognitive skills provide a better understanding of the environment in order to interact with it in the most efficient and safe manner. Some domains of cognitive functioning, such as attention, somehow decrease with mental illness. Some diseases contribute to a lack of concentration, and others cause a total inability to focus. However, each disease commonly contributes to a certain cognitive domain disruption. Lack of executive functioning is seen in many psychiatric conditions, but particular functions decrease to a certain degree. For instance, lack of planning is associated with anxiety disorders [1]. Autistic spectrum disorders involve a lack of flexibility [2], depressive disorders cause difficulties in decision-making and initiating activities [3]. Obsessive-compulsive disorder and bipolar disorder undermine an individual’s ability to inhibit poor responses [4], and schizophrenia causes a deficit of all the functions listed above [5]. Memory is affected in most psychiatric disorders. Declarative semantic memory loss is associated with schizophrenia, and while declarative episodic memory decreases in many psychiatric conditions, such as bipolar disorder and depression, it is especially prevalent in dementia and schizophrenia [3,5,6,7,8]. Social cognition (theory of mind and empathy) is severely damaged in schizophrenia, bipolar disorder [8], major depression [9], attention deficit hyperactivity syndrome [10], and obsessive-compulsive disorder [11,12] and is also common in autism, although to a lesser degree [6]. Impaired social cognition is tied closely to problems with understanding and decoding verbal speech, which is typical for mental disorders. Delayed verbal communication skills and a lack of prosody and pragmatics are characteristics of autism [13], while poor performance on verbal fluency tasks, semantic comprehension errors, and disorganized speech are common signs of schizophrenia [14]. Cognitive impairment, in varying degrees, accompanies all neuropsychiatric disorders. This review focuses on the neurobiological basis of cognitive decline in schizophrenia, depression, mild cognitive impairment (MCI), and dementia, particularly Alzheimer’s disease (AD).

Cognitive impairment is one of the main signs of schizophrenia, together with positive and negative symptoms. Various researchers estimate that dysfunction of working memory, attention, processing speed, visual and verbal learning, and significant deficits in reasoning, planning, abstract thinking, and decision-making are present in 62–98% of patients with schizophrenia [15,16,17,18,19].

Cognitive decline has not only been described in patients with ongoing schizophrenia [20], but also in periods of remission [21], in patients with a first psychotic episode [22,23], and even in healthy close relatives [23,24], and in persons with a high risk of developing the disease [25]. Researchers believe that neurocognitive impairment can be detectable long before the disease manifests and is just as significant as positive and negative symptoms [26]. The fact that cognitive decline is a precursor to significant functional impairment in patients with the final stage of schizophrenia [27] makes prompt diagnosis and treatment extremely important. 

Schizophrenia is characterized by cognitive distortions such as impaired concentration, thinking, the speed of cognitive information-processing, and verbal working memory. Cognitive symptoms persist throughout the disease and determine the functional status of patients [28]. Cognition takes place in different areas of the brain. Regulating mechanisms regarding learning ability and memory have been ascribed to the hippocampus, but there are still questions to answer with regard to other cognitive processes. 

Depressive disorders are also accompanied by cognitive impairment in various areas, such as attention, executive functions, memory, and processing speed [29,30]. These impairments often persist even during remission. Moreover, cognitive impairment is present in 85–94% of cases during exacerbations and 39–44% of cases during remissions [31]. Depressive disorders are a heterogeneous group of conditions in terms of severity, symptom composition, and treatment efficiency. They are widespread and affect 1.3–19% of people worldwide [32]. A variety of neuropathological mechanisms underlies this heterogeneity. An accurate and timely diagnosis could help prevent the risk of chronic conditions and the subsequent development of complications such as cardiovascular and cerebrovascular diseases and dementia [33,34,35]. Separate studies of genetic, proteomic, neurocirculatory, and clinical profiles have shown promising results but have not led to the development of a specific approach to diagnose and treat individual patients with depression. The determination of biological markers could help better diagnose depression and classify its subtypes, i.e., divide patients into more homogeneous groups that are clinically different in order to provide a personalized approach to each patient. Given the heterogeneity of depression, it is highly unlikely that a single biomarker could help predict the course of the disease or the treatment outcome. However, by evaluating a combination of parameters, using different laboratory and neurophysiological methods, it could be possible to create a list of biomarkers with significant predictive value [36,37,38].

MCI is a transitional stage between healthy aging and dementia. MCI affects approximately 10–15% people over the age of 65 [39]. There are two major subtypes of MCI: (1) amnestic MCI (aMCI) is MCI with memory loss as the predominant symptom, and (2) nonamnestic MCI (naMCI). Sometimes MCI is called «preclinical AD». However, not all patients develop dementia, and there is currently an active search for biomarkers that could be used to predict the development of dementia. The search for such biomarkers is an extremely important line of study.

Early diagnosis of MCI is important for many reasons. The most important is that the early detection of cognitive decline could allow intervention before further significant cognitive impairment occurs and delay or prevent the progression of dementia. Early diagnosis also gives individuals with the disease and their families enough time to plan ahead in case they are unable to make decisions in the future. Currently, there are no particular screening programs; however, early diagnosis of such disorders could promote early therapy and prevent the development of severe cognitive impairments, thereby reducing the economic load on the government and the burden on the healthcare system [40].

Dementia is a general term for a range of diseases and conditions that cause significant cognitive loss and decline in function. Almost all types of memory are impaired in neurodegenerative diseases, the main representative of which is AD. Much recent effort has been focused on the most detectable condition called mild cognitive impairment (MCI), conceptually defined as cognitive impairment that is more severe than in normal aging, but not severe enough to fit the criteria for dementia.

Alzheimer’s disease (AD) is recognized by the World Health Organization (WHO) as a public health priority worldwide. The progress of modern medicine has increased life expectancy around the globe. However, neurodegenerative diseases increase with age, too. For example, the percentage of people suffering from the most common neurodegenerative pathology, AD, at 65 to 74 years old is 3%, and at the ages of 75 to 84 years, it is 17%.

AD is a progressive disorder manifested by impaired memory, cognition, and behavioral functions that ultimately lead to mood swings and dementia. Despite great advances in understanding the pathogenesis of AD since Alois Alzheimer reported the first case in 1907, there are still no treatments to cure the disease. Typical late-stage AD likely comes from complex interactions between biological and environmental factors. It is now believed that ~70% of AD predisposition is due to genetic factors. Targeted genetic approaches and next-generation sequencing studies have identified a number of genes underlying a relatively high risk for AD, which also provides insight into disease pathogenesis [41].

The earliest clinical symptom of AD is usually impaired episodic memory [42,43,44,45]. Individuals with memory distortions (impairment of memory and recognition) and classified as pre-amnestic MCI and no underlying medical conditions that could explain these cognitive impairments (such as cerebrovascular disease, anxiety, stress or depression) have an 8.5-fold increased risk of progression to dementia (mainly AD) compared to those classified as possible-naMCI [46,47].

Psychiatric disorders are very heterogeneous, with a wide range of clinical and biological manifestations. One of the current areas of modern neurobiological investigations is the search for biomarkers of mental illness based on the known links in the pathogenesis of these diseases. Biomarkers are objective biological indicators associated with the presence, severity, prognosis, or response to a specific treatment of a particular disorder [48]. One of the main goals of biomarker research is to enhance diagnostic accuracy in order to improve patient outcomes. A potential biomarker should be determined in patients with mental disorders but not in healthy individuals. However, the fundamental definition of mental disorder is based on subjective clinical criteria. Therefore, a biomarker alone cannot be used to decide whether or not a person has a disorder [49]. 

Neurobiological mechanisms leading to cognitive impairment in various neuropsychiatric diseases include:Genetic mechanisms. This involves the dysfunction of various genes involved in the pathogenesis of mental disorders. Both genetic and environmental factors influence cognitive development, but while the genotype remains constant, epigenetic (environmental) factors may vary and influence cognitive functioning throughout a person’s lifetime and even before birth. For example, babies whose mothers drank alcohol during pregnancy can be born with fetal alcohol spectrum disorder (FASD) and have severe cognitive impairment [50]. Genomic studies have identified thousands of genetic loci associated with mental illness according to the results of multiple genome-wide association studies (GWAS) [51,52].Epigenetic mechanisms. Epigenetics is currently attracting more and more attention due to its role in mental illness studies, and it can provide a certain understanding of the causes both of schizophrenia and cognitive dysfunction. The understanding epigenetics of chromatin stability, gene regulation, response to environmental factors, and disease states has grown rapidly over the past two decades [53]. Epigenetic mechanisms tightly control gene expression and repression without any changes in the DNA sequence and, importantly, may serve as a mechanism to explain the absence of heredity in schizophrenia [54]. Epigenetics is a complex biological process that regulates DNA accessibility and gene transcription [55]. Chromatin is regulated by DNA methylation and post-translational modification of histones. DNA methylation is a process involving attachment of a methyl group to cytosine in DNA [56]. DNA methylation is catalyzed by related DNA methyltransferases (DNMT) [57,58]. These modifications alter gene transcription and can be highly stable and hereditary [59]. In contrast, histone modification is a more dynamic and complex process with a large number of post-translational modifications [60]. The regulation of histone tails is more easily reversible than DNA methylation and is likely to play an important role in neuroplasticity and disease pathogenesis [61].Dysfunction of neurotransmitter systems. This involves the impaired neuroplasticity and synthesis of neurotrophic factors that support nerve cell function The dysfunction of neurotransmitter systems, impaired neuroplasticity, and synthesis of neurotrophic factors that support nerve cell function are direct consequences of disorders in genetic and epigenetic mechanisms associated with cognitive decline. Discrete neural networks functioning by means of neurotransmitter systems underlie all cognitive processes. Dopamine, noradrenaline (norepinephrine), serotonin, acetylcholine, glutamate, and γ-aminobutyric acid (GABA) play an important role in the regulation of cognitive processes. Understanding neurobiological processes underlying cognitive functioning is essential for interpreting the behavior of both healthy and mentally ill persons [62].Neuroinflammation. Most psychiatric disorders, and consequently, cognitive dysfunctions, have an inflammatory component. Inflammatory processes accompany the disease, but some researchers believe that neuroinflammation may play a key role in the pathogenesis of the disease. Inflammatory processes also have a neuroprotective role, performing a protective function in the disruption of CNS structure and function [63]. Inflammation is seen as an integral part of the mechanisms of CNS homeostatic repair and defense [63]. The immune system is involved in many CNS processes, including neurodevelopment, synaptic plasticity, and circuit maintenance [64]. Thus, neuroinflammation is a multifaceted process. The process of neuroinflammation includes (1) induction of a local immune response by immune cells in the CNS, (2) higher production of pro-inflammatory cytokines and chemokines, (3) additional recruitment of immune cells from the CNS to the primary site of injury or infection, (4) permeability of the blood-encephalic barrier and leukocyte penetration, from blood to brain, and (5) resolving inflammation and tissue remodeling [64]. The role of neuroinflammation in the formation of psychiatric disorders is bidirectional. On the one hand, any influences, such as stress or neurodegeneration, lead to a disruption of homeostasis and neuroinflammation. On the other hand, chronic inflammation and chronic subclinical inflammation can lead to CNS dysfunction, specifically cognitive dysfunction.Vascular pathology. Many mental disorders are comorbid with cardiovascular diseases [65]. Cardiovascular risk factors (obesity, hypertension, diabetes, hypercholesterolemia, and smoking) can be associated with cognitive decline [66].The Framingham General Cardiovascular Risk Score (FGCRS) is a scale designed to assess cardiovascular risk factors. Scores on this scale are correlated with cognitive decline and Mini-mental State Examination scales (MMSE) [67], the Preclinical Alzheimer Cognitive Composite [68,69], a more rapid decline in memory, executive function, and verbal fluency [66]. Chronic pro-inflammatory status is also closely related to cardiovascular disease, which is also a pathogenetic link to cognitive decline in mental illness [65].Mitochondrial dysfunction. Mitochondria in eukaryotic cells act as an energy center; disruption of their function can lead to disruption of metabolic processes in the cell [70]. Mitochondria play an essential role in multiple neuronal functions: synaptic transmission, Ca^2+^ signaling, action potential generation, and ion homeostasis; impaired mitochondrial function contributes to impaired brain neuroplasticity [71]. Normal mitochondrial activity in the brain is also important because the brain uses large amounts of ATP but is unable to store large amounts of energy [70].

Mitochondrial dysfunction is seen in a variety of mental illnesses, including schizophrenia, depression, and neurodegenerative diseases.

Mitochondrial dysfunction leads to disruption of the following processes: changes in neurotransmitter systems, the generation of reactive oxygen and nitrogen forms, cell stress response, Ca^2+^ signaling, disturbances in glucose and lipid metabolism, changes in amyloid-beta (Aβ) metabolism, protein aggregation, disruption of cellular energy metabolism, and cell death [72]. All of the described processes contribute to cognitive impairment [73]. Polymorphisms in genes responsible for mitochondrial function are associated with mental illness.

In our review, we focused on neurobiological mechanisms and associated markers of cognitive impairment in various mental illnesses, schizophrenia, depression, MCI, and dementia. In each section, genetic, epigenetic, neurotransmitter, neurotrophic, neuroinflammatory, vascular, and mitochondrial aspects will be discussed. 

## 2. Schizophrenia

Cognitive function is to be genetically correlated with health-related issues and with neuropsychiatric disorders, in particular with the risk of developing schizophrenia [74,75]. GWAS identified 21 genomic loci that jointly influence cognitive function in schizophrenia. The strongest common locus was found at 22q13.2 (TCF20, CYP2D6, and NAGA). NAGA encodes lysosomal enzymes, and CYP2D6 encodes cytochrome P450 enzymes metabolizing a drugs. These genes could be potential drug targets for improving cognitive functioning in schizophrenia. PICK1 polymorphisms (rs3952 and rs2076369) are associated with cognitive decline in schizophrenia. A/A homozygotes rs3952 demonstrated better results in the tracing subtest. Patients with rs2076369 G/T genotype demonstrated better performance than homozygotes T/T in cognitive scores. G/G homozygotes performed better than T/T in the category fluency subtest [76]. The rs6984242 were demonstrated strongest associations both intelligence quotient (IQ) and episodic memory in schizophrenia [77]. The interleukin 10 (IL-10) allele could contribute to cognitive impairments in schizophrenia [78].

Epigenetic changes associated with schizophrenia were shown, as were some epigenetic mechanisms associated with cognitive impairment in schizophrenia. A neuroimaging study showed a decrease in histone deacetylase (HDAC) expression in the dorsolateral PFC correlating with cognitive impairment [79]. There was also a decrease in the level of HDAC2 messenger RNA (mRNA) in the dorsal lateral PFC of the patients [80]. It was shown that in mice, HDAC2 expression regulated the performance of cognitive tasks [81], and HDAC2 was enriched in the *N*-methyl-D-aspartate receptor (NMDAR) subunit promoter compared to HDAC1. HDAC2 knockout mice exhibited an increase in long-term potentiation (LTP) and enhanced performance on some cognitive tasks, which contradicted the results of the human study.

Several studies showed that the epigenetic regulation of NMDAR contributed to the processes of normal brain development and plasticity. In particular, histone methylation in gene promoters was associated with the regulation of development and the regionally specific expression of ionotropic and metabotropic glutamate receptors in the human brain [82]. Epigenetic regulation of NMDAR has also been observed in psychiatric diseases. Prenatal exposure to bisphenol A (a chemical used to make plastic for food containers) was associated with neurodevelopmental disorders, altering Grin2b methylation in mice and humans [83], which could mean that environmental influences altered the epigenetic regulation of NMDARs. Oxytocin methylation at the Chr3:8767638 site was associated with composite cognitive performance independent of demographic and medication factors [84].

The diagnosis of schizophrenia is associated with obvious changes in brain structure and neurotransmission. Two of the most influential hypotheses regarding the underlying neurobiology of the disorder consider glutamate and dopamine neurotransmitter systems.

Among the impairments underlying neurotransmission pathogenesis of cognitive impairments in schizophrenia, the glutamate system plays a central role. The authors believe that NMDAR hypofunction appears to be the starting point for the progression of schizophrenia symptoms, especially for cognitive impairment [85,86,87]. NMDARs are crucial both for synaptic plasticity connected to learning and memory and proper neuronal function. It is important to note that NMDARs are involved in the formation of cognitive patterns in the early postnatal stages of brain development, which are also called «critical development windows» [85,86,87].

Interneurons of GABA are the main inhibitory neurons in the CNS, playing a crucial role in various physiological processes. The GABAergic system is the main point of intersection of biological and environmental risk factors for schizophrenia [28].

Several markers associated with GABA neurotransmission appeared to be altered in the cortical areas of schizophrenic patients. A decrease was detected in the level of mRNA and protein for glutamic acid decarboxylase (GAD67). This enzyme is responsible for synthesizing most of the cytosolic and vesicular GABA in the dorsal lateral PFC, which is connected with working memory and selective attention [28]. Decreased GAD67 expression in schizophrenia is associated with the disinhibition of the cortical excitatory neurons, decreased neuronal oscillations, and synchronism [88].

The α5 subtype makes up 5–10% of the total number of GABA-A receptors in the brain; the hippocampus contains 25% of these receptors. A5-GABA receptors are mainly located in the dendritic regions CA1-CA3 of the hippocampus. Rodent studies show a role for these receptors in cognitive function [89].

Selective modulation of the GABAergic system is a is a prospective treatment strategy of cognitive impairment associated with schizophrenia. α5 subtype-selective ligands could be used as nootropic drugs [28]. Inhibitors of GAT-1, the main plasma membrane of the GABA transporter in the brain, helped overcome cognitive impairment in a lipopolysaccharide-treated rat model [90]. A variety of GABAB receptor antagonists displayed cognition-enhancing effects in animal models of psychiatric disorders [28]. A high heritability of schizophrenia of approximately 80% indicates a strong genetic component of the disease [91]. Relatives or siblings of schizophrenic patients also have cognitive impairment, but to a lesser degree. These data provide evidence for the contribution of genetic factors to cognitive impairment in schizophrenia [74]. The onset of schizophrenia is likely due to a complex interaction between environmental risk factors and genetic predisposition [92]. 

Neuroplasticity and neurogenesis are adaptive functions of the brain. Violation of these processes manifests in the form of various mental illnesses. In schizophrenia, impaired neuroplasticity is specifically associated with disrupted cognitive functions. Brain-derived neurotrophic factor (BDNF) is a protein in the growth factor family, the subfamily of neurotrophins; it is detected in glial, but mainly in neuronal cells. BDNF is a neurotrophin that promotes the survival of existing neurons and stimulates the growth and differentiation of new neurons and synapses. It is the most studied protein of this class [93]. BDNF plays a role in the processes of dendritic spines and outgrowths formation, acting as a true regulator of axon growth. It is suggested that learning and memorization procedures cause synapse strengthening effects, resulting in long-term potentiation [94]. Meta-analysis showed a correlation between a decrease in BDNF and cognitive functioning in schizophrenia. In meta-analyses of cognitive domains, BDNF levels were significantly associated with verbal memory, working memory, processing speed, and verbal fluency performance [93]. However, the same authors showed that decreased BDNF levels did not play a major role in cognitive dysfunction in most patients with schizophrenia.

The neuroinflammation theory is a popular theory that seeks to explain the pathogenesis of mental disorders. The association between increased levels of inflammatory markers and cognitive decline in inflammation-related conditions suggested a role for inflammation in the regulation of cognition under physiological conditions [95]. The role of inflammation mediators in learning and memory was previously discussed under the name “cytokine model of cognitive function”. Microglia plays a role in the immune function of the brain and regulates cognitive function under physiological conditions. Depletion or inhibition of microglia in mice negatively affected learning and memory [96]. Microglia can also produce pro-inflammatory cytokines. For example, IL-1β, IL-6, and tumor necrosis factor (TNF) -α are necessary for the physiological regulation of memory processes since disruption of their signaling pathways leads to a decrease in learning and memory, but their overexpression disrupts normal learning and memory systems [95]. IL-1, IL-6 and TNF may play a role in maintaining synaptic plasticity, neurogenesis and mechanisms involved in learning, memory and cognition [94]. Microglia interact with synapses in a glutamate-dependent manner, which suggests their role in learning and memory through their influence on synaptic plasticity. Anti-inflammatory cytokines such as IL-10 are also involved in the regulation of synaptic plasticity in the hippocampus in a dose-dependent manner. Microglia and inflammatory mediators may also indirectly influence biological mechanisms associated with cognition through modulation of neurotrophic factor levels and activation of signaling pathways. Cytokines such as IL-1β and IL-6 play a dual role in adult neurogenesis in the hippocampus. Not long ago, IL-10 was described as an enhancer of postnatal neurogenesis [97]. There is evidence that cytokines can modulate the concentration level and activity of BDNF. Immune stimulation decreases the expression and activity of BDNF in the brain, which leads to altered synaptic plasticity in the hippocampus [98]. Inflammatory processes can influence the activation of the kynurenine pathway. Pro-inflammatory cytokines cause the activation of the kynurenine-producing enzyme indoleamine 2,3-dioxygenase (IDO) in the hippocampus, which is involved in the regulation of learning and memory [99]. Dysfunction of the kynurenine pathway, mediated by inflammation, may also be involved in cognitive changes in schizophrenia, as patients have increased levels of kynurenine in the *cerebrospinal fluid* (CSF) [100]. This effect may be mediated by changes in glutamatergic neurotransmission. Inflammation can promote the production of the NMDAR antagonist KYNA, causing glutamatergic imbalance. Decreased glutamatergic neurotransmission may be associated with altered transport of synaptic vesicles associated with the concentration level of IL-10 in the brain of patients with schizophrenia [95]. The role of inflammation in cognitive impairment in schizophrenia was shown in animal models. For example, a pregnant female was exposed to polyinositol, which caused an inflammatory response. The offspring born were cognitively impaired and also had schizophrenic-like behavior. Prenatal maternal stress was linked to the development of schizophrenia in offspring [101]. An elevated CRP level was associated with cognitive impairment in schizophrenia [93]. Elevated CRP and pro-inflammatory cytokine levels were also associated with cognitive dysfunction in people with schizophrenia and bipolar disorder [102]. Associations were shown between general cognitive function and serum IL-6, sTNF-R1, and IL-1ra levels [103]; elevated peripheral IL-1β mRNA levels were associated with impairments in verbal fluency and brain volume reduction in patients with schizophrenia [104]. The association between IL-6 levels and cognitive performance depended on age and antipsychotic dose [105]. IL-1 and 6 (IL-1, IL-6) and TNF were linked to cognitive deterioration [106].

Influencing neuroinflammation processes could be an effective therapeutic strategy for treating cognitive impairment in patients with schizophrenia. One of the therapeutic options could be, for example, the inhibition of microglial activation [107,108]. For example, minocycline treatment contributed to the improvement of cognitive impairment in patients with schizophrenia. This effect is probably related to a decrease in proinflammatory cytokines by inhibiting microglia [109,110]; improvement in cognition factors in patients with schizophrenia was observed when the anti-inflammatory drug was added to Risperidone treatment.

The role of oxidative stress for brain cells and their functioning is also known. The results of one study [111] showed that the accumulation in brain cells of a high content of f-SATIII repeats could seriously change the normal functional activity of cells of various brain structures in people with schizophrenia. It was shown that the large size of rDNA clusters stabilized heterochromatin 1q12, reducing the intensity of satellite transcription, which contributed to an increase in the content of f-SATIII. Chronic oxidative stress induced an adaptive response only in cells with a low content of f-SATIII. Cells with a high content of f-SATIII, in which the adaptive response is blocked, were less resistant to damaging effects, and died. The low content of f-SATIII repeats in patients with schizophrenia may be a consequence of chronic oxidative stress and many copies of ribosomal repeats, which may also indicate a significant role in the inflammatory process of brain cells and the course of schizophrenia.

The association of vascular pathology, cognitive impairment, and schizophrenia has been shown. One of the mechanisms of the pathophysiology of schizophrenia is disruption of the brain’s microvascular system [112]. Brain hypoperfusion and microvascular abnormalities have been shown in various neuroimaging studies [112] and postmortem studies [113]. Antipsychotics’ D2-receptor antagonists can affect the neurovascular system and small vessels in the frontal cortex by altering the vasoconstriction–vasodilation balance, reducing blood flow and metabolism and causing structural microvascular changes proportional to the level of apoptosis at this stage. A postmortem study showed microvascular pathology in the frontal cortex of patients with schizophrenia or schizophrenic spectrum disorders treated with D2-blocking antipsychotics; one stage with functional, reversible changes that may correlate with impairment of working memory and the presence of extrapyramidal symptoms, and an irreversible stage with significant impairment of cognition and global functioning [113]. Inflammatory processes also disrupt the regulation of cerebral blood flow. Also, there is the role of vascular growth factors in the formation of normal brain perfusion. The role of vascular endothelial growth factor (VEGF) in the pathogenesis of cognitive disorders in schizophrenia is the most studied. VEGF has multiple functions; it participates in angiogenesis, protects against brain cell loss, blood–brain barrier (BBB) dysfunction, dendritic spine loss, regulates blood flow, and causes vasopermeability of vascular endothelial cells. VEGF is also involved as a neurotrophic factor in brain homeostasis [114]. Impaired VEGF expression may underlie the hypoperfusion or decreased blood flow observed in patients with schizophrenia [115]. VEGF modulates hippocampal synaptic plasticity associated with memory function [116,117]. VEGF was elevated in the parietal cortex of patients with schizophrenia compared to controls, and serum VEGF levels were associated with PFC volume in schizophrenic patients [118]. Several data suggest that altered cerebral circulation due to changes in the VEGF system affects cognitive ability in patients with schizophrenia [114]. VEGF levels inversely correlate with the severity of cognitive impairment [114]. A study [119] determined a vascular endothelial index (VEI) based on serum levels of VEGF, intercellular adhesion molecule-1 (ICAM-1), and vascular cell adhesion molecule-1 (VCAM-1). Cell adhesion factors are important for the formation of the blood–encephalic barrier and normal perfusion of brain tissue. A linear combination of ICAM-1 and VCAM-1 levels was most different from healthy controls, with significantly higher levels of this composite VEI in patients with schizophrenia than controls. Patients with schizophrenia with higher VEI had an earlier age of onset [119]. In a study by Zhao (2019), remitted first-episode schizophrenic patients had deficits in neurocognition but not social cognition; these deficits were associated with serum VEGF levels. As the severity of cognitive impairment in schizophrenia decreased, VEGF levels gradually decreased. VEGF may be involved in the mechanisms underlying cognitive function in patients with remitting schizophrenia, although the mechanism of this relationship remains poorly understood. The authors believe that VEGF may serve as a sensitive monitor for assessing the degree of cognitive impairment and clinical prognosis in schizophrenia [115]. 

We suggest that VEGF may be a marker of cognitive impairment in schizophrenia, but it cannot be an independent marker because its impairment is not specific and does not occur only in CNS diseases. However, it is possible to use this marker in a comprehensive assessment.

The role of mitochondrial dysfunction in the incidence of schizophrenia has been shown [71]. Within the framework of our review, we will describe only those possible disorders observed in the cognitive impairment in schizophrenia. There are not many studies available.

Hemizygous deletion of a 1.5- to 3-megabase region on chromosome 22 causes 22q11.2 deletion syndrome (22q11DS), which constitutes one of the strongest genetic risks for schizophrenia. Mice with this deletion showed haploinsufficiency of Mrpl40 (mitochondrial protein 40 of the large ribosomal subunit) and abnormal short-term synaptic plasticity that contributes to working-memory deficiencies similar to those in schizophrenia. The authors suggest that mitochondrial calcium deregulation is a novel pathogenic mechanism of cognitive impairment in schizophrenia [120]. Transgenic mice G72Tg that carry the human-specific mitochondrial gene locus G72/G30 harbor reduced activity of mitochondrial complex I and showed schizophrenia-like behaviors, which can be rescued by pharmacological treatment with the glutathione precursor N-acetyl cysteine [71].

A study [121] in a Mexican population shows that circulating cell-free mitochondrial DNA (cf-mtDNA) fragments in blood plasma may be a potential biomarker for determining the cognitive status of patients with schizophrenia. Cognitive function was assessed using the Montreal Cognitive Assessment (MoCA) scale.

Schizophrenia is a highly multifactorial disease characterized by a wide variety of clinical manifestations. Because each case is unique and differs from others in its symptoms, it is difficult to find adequate and valid prognostic and therapeutic biomarkers to use. However, genetic, epigenetic, neurotrophic, inflammatory, and oxidative biological correlates of cognitive decline may help in understanding the pathogenesis of this disease and open up possibilities for effective treatment.

## 3. Depression

The two main types of cognitive dysfunction observed in depressive patients are cognitive biases, which include altered information processing or sustained attention towards negative stimuli, and cognitive deficit, which include impairments in attention, short-term memory, and executive function. Lack of cognitive functioning, attentional bias, and persistent negative affect are mainly due to dysfunctions of prefrontal–subcortical circuits and associated impairments in cognitive control of emotions. Numerous studies have shown a decrease in metabolic activity or resting blood flow in the dorsal regions of the PFC, together with increasing ventral cortical and limbic activity in depressive disorders. Overactivity of the ventral PFC and limbic regions could mediate negative stimulus assessment and neurovegetative, motor, and neuroendocrine aspects of depression, while inadequate activity in the dorsal regions of the PFC and anterior cingulate cortex could mediate attention and working memory deficit and impairments in cognitive control of emotions. The combination of these changes contributes to an imbalance between the two systems with an intermittent prevalence of disorders in one or another part of the brain, which partly explains the observed clinical heterogeneity of depression [122].

Some genetic markers of cognitive dysfunction have been found. For example, the STin2.10 allele in the serotonin transporter gene is twice as common in depressed patients compared to healthy individuals [123]; its carriers have lower executive functions. In another genetic study, carriers of the Val66Met single nucleotide polymorphism (SNP) in the BDNF gene showed signs of impaired episodic memory [124], which was associated with decreased hippocampal activity on functional *magnetic resonance imaging* (MRI) [125]. The lack of reliable and significant genomic variation between depressed and clinically healthy populations [126] resulted in the development of new genetic approaches, such as identifying polygenic factors that, together with adverse environmental conditions, influence the development of depressive diseases [127] or comparison studies of telomere length that play an important role in the preservation of genome information. It is known that the rate of telomere shortening increases in different pathological conditions. Telomere shortenings were found in autism spectrum disorders, schizophrenia, AD, Parkinson’s disease, and depression, and several studies showed a link between telomere shortening and disease severity [128]. GWAS analysis was performed with older depressed adults to identify genetic variations associated with baseline and changes in the CERAD Total Score (CERAD-TS) in Neurocognitive Outcomes of Depression in the Elderly (NCODE). GWAS of cognitive function identified promising candidate genes that could be potential biomarkers for cognitive decline. The GWAS of baseline CERAD-TS revealed a significant association with an intergenic SNP on chromosome 6, rs17662598. The most significant SNP that lies within a gene was rs11666579 in SLC27A1. SLC27A1 was involved with processing docosahexaenoic acid (DHA), an endogenous neuroprotective compound in the brain. Decreased levels of DHA were associated with the development of AD. The most significant SNP associated with CERAD-TS decline over time was rs73240021 in GRXCR1, a gene previously linked to deafness [129]. 

miRNAs are a class of small non-coding RNAs that regulate synaptic plasticity and play an important role in depression pathogenesis. Peripheral miRNAs can be regarded as potential biomarkers for diagnosis and treatment response evaluation. Despite this, many unstudied questions remain, such as whether the brain is the source of peripheral miRNAs, providing that only some changes in blood miRNAs were confirmed by different studies [130]. As for the brain neurotrophic factor, seven different miRNAs were identified by genome sequencing in different parts of the human brain. They regulated the expression and functional activity of the factor [131]. Nevertheless, there is no doubt that additional studies of miRNAs are needed before they can be used as reliable markers of cognitive dysfunctions in depressive diseases.

A variety of neurotrophic factors that play an important role in the development of depression could also be markers of cognitive dysfunctions. It is well known that one of the molecular factors required to maintain neuroplasticity is BDNF. Mature BDNF (mBDNF) and its TrkB receptors are found not only in nerve tissue cells but also in endothelial cells, cardiomyocytes, vascular smooth muscle cells, leukocytes, and megakaryocytes. It was shown that the synthesis of mBDNF occurs in the precursors of platelets—megakaryocytes. Platelets bind, store, and release mBDNF when activated at the site of traumatic injury, helping to repair peripheral nerves or other TrkB-containing tissues. However, there is no scientific evidence that changes in serum mBDNF levels are directly related to changes in mBDNF levels in the brain and neuroplasticity processes. There are contradictory data on changes in the level of mBDNF in platelets and serum in patients with depression [132]. Studies of the postmortal tissues of the PFC and hippocampus of suicide victims revealed a significant decrease in the expression of BDNF and its TrkB receptors compared to healthy people [133,134]. Although there are reasons to evaluate this marker, the issue requires additional research. Precursor BDNF (proBDNF) is understudied compared to mBDNF. It is known that proBDNF mostly interacts with the low-affinity neurotrophin receptor p75 (p75NTR), and it slows down or even stops the growth of axons and branching of neuronal dendrites, which leads to a decrease in the number of synapses. Such a low functional activity of mBDNF due to a decrease in its expression might presumably lead to the predominance of proBDNF effects. According to recent studies, proBDNF could be elevated in patients with depression [135].

Analysis of nerve growth factor (NGF) in peripheral blood showed its decrease in patients with depression compared to healthy controls. There were no changes in its level after treatment with antidepressants, even though patients with severe depression had the lowest level of NGF [136]. Similar results were found in studies of glial cell-derived neurotrophic factor (GDNF) [137]. VEGF plays an important role in stimulating angiogenesis and neurogenesis and appears to be a promising marker [138]. Studies found an increased level of VEGF in the blood of patients with depression compared with controls [139,140]. Studies of the effect of insulin-like factor-1 (IGF-1) and basic fibroblast growth factor (FGF-1, 2) in patients with depression found that their levels were also increased compared to healthy controls. However, research data are inconsistent and require further study [141,142]. The neuronal thread protein AD7c-NTP is closely linked with the tau protein, which is elevated in the CSF and urine of AD patients. AD7c-NTP is a transmembrane phosphoprotein, which is identified in CNS cells. The over-expression of AD7c-NTP could lead to neuritic sprouting and cell death, which is related to pathological changes in AD. The accumulation of the tau protein is associated with both AD and late-life depression. The urinary levels of AD7c-NTP in late-life depression with the cognitive impairment group were significantly higher than both the late-life depression without cognitive impairment and healthy control groups but lower than in the AD group. The level of urinary AD7c-NTP appears to be associated with cognitive impairment in late-life depression [143].

Cognitive functions in depression are influenced by elevated levels of inflammatory mediators and increased inflammatory responses caused by altered activity of enzymes in the main metabolic pathways of neurotransmitters (decreased tryptophan levels and increased tryptophan metabolites) [144]. Also, an increased level of inflammatory cytokines alters the level of monoamines. Neuroinflammation, manifesting in microglia activation, leads to increased oxidative stress, pathological reduction in the number of synapses, and impaired neuroplasticity. The developing inflammatory response also activates the hypothalamic-pituitary-adrenal system, which leads to hypercortisolemia and metabolic disorders, further contributing to neuronal dysfunction [145].

The most significant changes were found for two pro-inflammatory factors: IL-6 and CRP, which were increased in patients with depression compared to healthy controls [146]. At the same time, it was found that a high level of CRP was associated more with impaired cognitive parameters (low psychomotor speed) than with the main symptoms of depression [147]. Higher serum CRP levels in women were significantly correlated with Montgomery Aasberg Depression Rating Scale symptom severity, severity on observed mood, cognitive symptoms, interest-activity, and suicidality [148]. The prospective cohort study assessed serum CRP, IL-6, and cognitive symptoms of depression over a long period of time, averaging 11.8 years. CRP and IL-6 at the first point of follow-up predicted cognitive symptoms of depression at follow-up, while baseline symptoms of depression did not predict inflammatory markers [149]. IL-4, IL-5, IL-12, IL-13, IFN-γ, and TNF-α significantly negatively correlated with the Beck Depression Inventory (BDI-II) cognitive factor [150]. A study that included repetitive transcranial magnetic stimulation (rTMS) treatment of patients with treatment-resistant depression demonstrated that partial improvement in cognitive dysfunction was observed after transcranial magnetic stimulation; the authors suggest that this may be due to a decrease in peripheral IL-1β levels [151]. According to numerous studies, changes in inflammation indicators could be used as prognostic markers of the selected therapy’s effectiveness (IL-2, IL-4, IL-8, IL-10, and IF-gamma) [152,153].

High comorbidity between depression and cardiovascular diseases has been shown. The pathological mechanisms include excessive activation of the hypothalamic-pituitary-adrenal axis and dysregulation of the immune system, which causes chronic pro-inflammatory status [65].

The term vascular depression exists in the literature. There are relationships between late-life depression, vascular risk factors, and cerebral hyperintensities, the radiological sign of vascular depression [154]. Cognitive dysfunction is common in late-life depression, especially executive function dysfunction, which is an indicator of poor response to antidepressants. Over time, the progression of hyperintensity and cognitive impairment predicts a severe course of depression and may indicate an underlying impairment of vascular disease [154].

When there is an energy imbalance, the body becomes more “vulnerable” and more susceptible to stress-related disorders such as depression [70]. It has also been hypothesized that there is a neurobiological link between inflammation, oxidative stress, and treatment resistance in depression [70]. Depression may be caused by impaired energy metabolism in the brain due to mitochondrial genetic vulnerability (A3243G mutation, mitochondrial RNALeu(UUR) transfer) [70]. Reactive oxygen species (ROS) and reactive nitrogen species (RNS) are involved in many homeostatic processes, including the cellular response to stress, modulation of autophagy, mitochondrial function, signaling, and apoptosis [155]. The main producers of ROS and RNS in the cell are mitochondria [155]. The brain is also the most vulnerable organ for reactive forms [70]. ROS and RNS under certain conditions can cause damage to membrane phospholipids, sugars, DNA, and proteins, thereby negatively affecting cellular function [155].

When the balance between pro-and antioxidants is disturbed, oxidative stress occurs. Oxidative stress is also closely linked to neuroinflammatory processes [156]. All of these play key roles in the development of depression. 

Regarding cognitive impairment, oxidative stress has been associated with cognitive impairment in animal models [157]. ROS and RNS play an important role in the progression of cognitive dysfunction in recurrent depression [158]. Some areas of the human brain contain significant amounts of metal ions which contribute to ROS. Furthermore, lower concentrations of antioxidants are observed in CNS tissues compared to other organs. Cells of the CA1 area of the hippocampus (Sommer sector) and CA4 (Bratz sector), cells of dorsal-lateral striatum, and neurons of layers III and V of the cerebral cortex are considered to be the most sensitive to damage. A decrease in visual-spatial and auditory-verbal working memory span and declarative memory is associated with elevated levels of malondialdehyde, an indicator of antioxidant system efficiency and damage caused by reactive oxygen species, increased levels of nitric oxide (NO), and reduced levels of antioxidant protection. Patients with major depression have decreased plasma ability to absorb oxygen radicals compared to healthy individuals [158].

ROS, RNS, and inflammation play a common role in the pathopsychology of depression and may also explain the link between depression and neurodegeneration [156]. There is also the hypothesis that reactive forms play a central role in the onset of rapid aging in depression [156].

Thus, markers of neuroinflammation (IL-6, C-reactive protein), neurotrophic factors (BDNF, VEGF), various genetic variations and epigenetic dimension, as well as measured levels of oxidative stress, can be used together as markers of cognitive dysfunctions in depressive disorders; therefore, further research needs to be conducted.

## 4. MCI and Alzheimer’s Disease

The complex neuropathology of dementia, and its early stage, MCI, includes many causes, including the synaptic atrophy and synaptic loss, neurovascular pathology, alterations in the innate-immune response, the accumulation and aggregation of hyper-phosphorylated tau proteins into neurofibrillary tangles that disrupt the normal neural cell cytoarchitecture, accumulation of tau-protein and Aβ in various brain structure [159]. These characteristics collectively underscore the involvement of multiple pathogenic pathways. It is important to note that the value of each of these neuropathological biomarkers varies extensively among prodromal, moderate, and severe AD conditions, and none of these numerous signs of AD changes is characteristic or distinctive for the AD phenotype. In other words, many of these signs are partly typical of other age-related neurological diseases of the human CNS. We will consider several factors that could be biomarkers for AD, MCI, and the progression of MCI to dementia.

The standard for diagnosing AD is the presence of Aβ and tau proteins in the CSF [160]. Researchers suggest the use of Aβ precursors and the Aβ1-40/Aβ1-42 ratio as biomarkers of AD. However, other studies have indicated that these markers might not be reliable in different populations [161]. The evaluation of Aβ plasma concentrations also could not be a reliable marker of MCI progression in AD. The plasma levels of Aβ42 and Aβ40 were decreased only in AD dementia. In amyloid positive controls, the plasma concentrations of Aβ42 were mildly decreased, whereas the Aβ40 levels remained unchanged [162]. Amyloid precursor protein (APP) processing, sAPPβ and β-secretase activity were not useful diagnostic or staging markers in preclinical AD [163].

Blood p-tau181 could predict tau and Aβ pathologies and differentiate AD from other neuropathology [164]. Plasma p-tau181 was elevated in preclinical AD and further increased at the MCI and dementia stages [165]. Serum tau protein concentration correlated with cognitive scale scores. Thus, serum t-tau, but not p-tau, was significantly higher in the mild AD group compared to the control group, and there were significant correlations of serum t-tau with the MMSE, Clinical Dementia Rating (CDR), and Global Deterioration Scale (GDS) [166] scores. Elevated tau levels could be detected in MCI, especially in the entorhinal cortex, using positron emission tomography [39].

Thus, Aβ and tau proteins can serve as markers of cognitive impairment; however, they are insufficient and cannot detect all cases of dementia, especially in the early stages.

Genetic factors are of great importance as biomarkers of AD. The most investigated, but not the only, genetic risk factor for AD is the presence in an individual of the ε4 allele of the apolipoprotein E gene (APOE). APOE is the main constituent of plasma lipoproteins; it is also involved in their production, conversion, and clearance. APOE is also involved in the hepatic synthesis of apolipoproteins and their absorption by peripheral tissues, providing triglyceride delivery and energy storage in muscles, heart, and adipose tissues. It also participates in the lipoprotein-mediated distribution of lipids between tissues and plays an important role in lipid homeostasis in tissues. APOE plays an important role in lipid transport in the CNS, regulating neuronal survival and maturation. The ε4 allele occurs in 20–25% of patients with AD and increases the risk of developing the disease threefold in heterozygous carriers and 15-fold in homozygous carriers [167].

The efforts of many scientific research groups have been directed towards a detailed investigation into the underlying genetics of AD. Conducting large-scale genome-wide association studies, performed using samples of tens of thousands of AD patients and healthy donors, required the creation of international consortia such as the International Genomics of Alzheimer’s Project (IGAP), Consortium on the genetics of Alzheimer’s disease (Alzheimer’s Disease Genetics Consortium, ADGC), the Alzheimer’s Disease Neuroimaging Initiative (ADNI), and others. Research groups from the USA, England, France, Holland, Iceland and other countries participated.

Of all the organizations actively involved in the study of AD genetics, the most prominent are the University of California, USA (Departments of Neurology, Radiology and Biomedical Imaging, Cognitive sciences, Neuroscience, etc.) [168,169] and Cardiff University, UK [170,171]. As a result of joint research work, a large amount of data on the genetics of AD has been accumulated [172,173], and more than 40 loci associated with the disease were identified [174]. The collected data has made a significant contribution to understanding the pathogenesis and polygenic nature of AD, established a foundation for further research into the molecular processes underlying the development of the disease, and elaborated new diagnostic and therapeutic strategies. Nevertheless, individual SNPs at the identified loci, as a rule, had a small effect on the risk of developing the disease and could not be used as independent prognostic markers.

Based on the GWAS data, different research teams proposed polygenic models for the risk of AD development based on an assessment of the total (multiplicative) effect of several SNPs. Some examples of genetic associations included genes such as the APP gene encoding an outer cell membrane receptor protein and complement receptor type 1. This gene is a member of the complement activation regulator family. It was demonstrated that CR1 could act as a negative regulator of the complement cascade, mediate immune adhesion and phagocytosis, and also act as an inhibitor of both the classical and alternative pathways of cellular immunity. The EPHA1 gene codes a tyrosine kinase receptor. When activated, EPHA 1 induces cell attachment to the extracellular matrix, inhibiting cell movement and motility. It also plays a role in angiogenesis and regulates cell proliferation. It may play a role in apoptosis too. The gene CHRNA7 alpha-7 subunit of the neuronal acetylcholine receptor is associated with the efficacy of acetylcholinesterase inhibitors [175].

The Espinosa research group (2018) [47] examined correlations between known AD-associated SNPs and individual neurocognitive endophenotypes of people with MCI. The researchers identified the following neurocognitive endophenotypes in patients with aMCI: attention and working memory, processing speed and executive functions, verbal learning and memory and global cognition. In conducted analysis, apart from APOE, most of the other loci were not associated with neurocognitive endophenotypes. The APOE-ε4 allele demonstrated a significant difference in distribution with the following ordering between subtypes: probable-aMCI > possible-aMCI > possible-naMCI > probable-naMCI. APOE-ε4 appeared to have a significant association with performances in delayed recall in the aMCI group, while in the possible-aMCI group, there was an association between APOE-ε4 and performances in verbal learning. Also in the probable-aMCI group, there was a significant association between AP2A2 and repetition and HS3ST1 and backward digits. The authors suggested that such a small number of associations might be due to the fact that the currently identified AD risk loci only had a minor effect on neurocognitive endophenotypes of individuals with MCI. If this was the case, then the real effects of the selected loci could only be detected by studying large series or by means of large meta-analysis [47].

The pharmacogenetics of drugs used to treat AD challenged many research teams in Switzerland [176,177], Italy [178,179,180], China [181,182], and other countries. A number of studies showed a correlation between the effectiveness of therapy with cholinesterase inhibitors and markers in the genes encoding CHRNA7 (alpha-7 subunit of the neuronal acetylcholine receptor), CHAT (choline acetyltransferase), BCHE (butyrylcholinesterase), ACHE (acetylcholinesterase binding), and ABCA APOE [183]. However, the results in some cases were contradictory. For this reason, further studies on larger samples are required.

An important task is to find genetic biomarkers predicting the probability of progression from MCI to AD. Chaudhury (2019) created a polygenic model to better estimate the risk of transition from MCI to AD by observing patients for 36 months. The accuracy was 61.0% [184].

Some genetic variations are associated with AD and MCI and can be used as markers of MCI progression. However, there are many genetic variations, and their effect is small; more studies are needed on large samples of patients and healthy controls to create polygenic risk models with a high predictive ability for further clinical use.

Epigenetic mechanisms regulating gene expression include DNA methylation, histone acetylation, and miRNA. All three mechanisms are involved in the development of AD and MCI.

Methylation of the ADRB2 gene encoding beta-2-adrenergic receptor is associated with AD. Analysis of 120 Swedish pairs of twins showed that this specific DNA methylation sign is not predictive of AD as methylation occurs after the onset of the disease. Studies in mice showed that suppression of ADRB2 caused learning and memory problems, which are the main symptoms of AD [185].

Mancera-Páez (2019) [186] examined DNA methylation in the APOE gene and apolipoprotein E (ApoE) plasma levels in MCI. Increased plasma ApoE and APOE methylation of CpGs 165, 190, and 198 were risk factors for MCI. Investigation of the associations between common SNPs in genes regulating DNA methylation showed that rs1187120 SNP in DNMT3A moderated cognitive decline in subjects with MCI [187]. The other study did not confirm those associations [188]. DNA methylation of the SORL1 5’-flanking region could be associated with the manifestation of MCI with type 2 diabetes mellitus [189]. Sung (2016) [190] observed lower methylation of heme oxygenase-1 at the -374 promoter CpG site in AD patients compared to MCI and control, and a correlation between MMSE score and demethylation level. ATP Binding Cassette Subfamily A Member 2 mRNA expression was upregulated in AD compared with controls. Methylation of 2 of 36 CpG islands in the ABCA2 gene was negatively associated with AD risk [191].

Mahady (2018) [192] estimated the level of epigenetic proteins in frontal cortex HDAC and sirtuin (SIRT) levels in tissue obtained from subjects with a premortem diagnosis of no-cognitive impairment, MCI and AD of varying degrees of intensity. HDAC4 levels increased significantly during disease progression, while SIRT1 decreased in neurodegeneration compared to no-cognitive impairment group. HDAC1 levels negatively correlated with perceptual speed, while SIRT1 positively correlated with perceptual speed, episodic memory, global cognitive score, and MMSE. The authors suggested that altered epigenetic protein function is associated with the development of neural impairment and cognitive dysfunction in dementia [192].

Histone H4 is one of the five major histone proteins responsible for chromatin structuring in eukaryotes; H4K16 is the lysine amino acid residue at its N-terminus. The histone acetylated at this residue is H4K16ac. Changes in the location of this histone in the genome of a person with the disease can be divided into three classes: age-related, normal; age-related, pathological; and specific for AD [193].

Chromatin remodeling plays a role in AD mRNA alterations. BACE1 mRNA levels were increased in peripheral blood mononuclear cells in patients with AD, along with increased promoter accessibility and histone H3 acetylation [97].

miRNA, or regulatory RNA ~19–23 nucleotide (nt) single-stranded non-coding RNA (sncRNA), are important epigenetic, post-transcriptional regulators of mRNA complexity [159]. Swarbrick et al. (2019) [194] showed that at least 11 different miRNAs are disrupted in the early stages of AD, and also that such alterations are observed many years before the onset of clinical symptoms. The genes regulated by these miRNAs are responsible for the cell cycle and cellular response to stress, cell surface receptor (Wnt/β-catenin) signaling, and gene expression regulation [195]. Another study identified five miRNAs associated with the genes AGER, LINC00483, GPER1, and PHLPP2, which had differential expression in AD [196]. AGER (also known as RAGE) is significantly associated with the progression of AD through its effects on the inflammatory pathway, induction of oxidative stress, production and accumulation of beta-amyloid, impaired synaptic transmission, and neuronal degeneration. GPER1 is overexpressed in MCI. LINC00482 is a long intergenic non-coding RNA (lincRNA),also known as C17orf55; the function of this gene is currently unknown [197].

Piscopo (2019) performed genome-wide DNA methylation analysis in order to identify epigenetic imbalance in the blood of MCI patients. There was a correlation found between DNA methylation and transcriptome changes, which could be a biomarker of MCI [198]. Significantly, to date, not a single miRNA has been found that would make it possible either to diagnose the “prodromal” or “MCI” phase of AD or any particular stage of the disease. Recent results indicated that it was highly unlikely that any single miRNA in brain tissue, CSF, serum, urine, or any other body fluid from different populations of people could ever predict AD at any stage of the disease [159]. Nevertheless, it is essential to create an integrated approach to diagnose and select therapeutic strategies based on several identified biomarkers. Due to the large number and variety of miRNAs being studied, they could give us much useful information and deepen knowledge about the pathogenesis of neurodegenerative diseases. Dysfunction of neurotrophic factors can also be observed in AD.

BDNF is the most studied neurotrophic factor, and its decrease in the brain and/or blood plasma has been found in many psychopathologies. Although meta-analysis data showed a BDNF decrease in AD [199] and its concentration was associated with a reduction in hippocampal volume [200], other researchers believed that the possible biased use of plasma BDNF in MCI was critically risky [201]. Xiao (2019) [202] proposed the dipeptidyl peptidase-4 (DPP4) activity to BDNF ratio as a plasma biomarker. The Montreal Cognitive Assessment score decreased progressively when this ratio increased [203].

The role of another neurotrophic factor, neurotrophin NGF, was shown in the pathogenesis of AD too. β-NGF and its receptors, tropomyosin receptor kinase A (TrKA) and p75NTR, produced several biological responses, including cell apoptosis and survival, and inflammation [204]. Profound and early basal forebrain cholinergic neuron (BFCN) degeneration was a hallmark of AD. Loss of synapses between the basal forebrain and hippocampal and cortical target tissue correlated highly with the degree of dementia and was thought to be a major contributor to memory loss. BFCNs depended on their survival, connectivity and function on the neurotrophin NGF, which was retrogradely transported from its sites of synthesis in the cortex and hippocampus [205]. Synaptic damage, axonal neurodegeneration, and neuroinflammation were common features in dementias. CSF neurofilament light and neurogranin (biomarkers of synapse and neurodegeneration) differentiated health control subjects from neurodegenerative dementias [206].

There was a well-established association between neuroinflammation and neurodegenerative diseases. Also, activation of the immune system could contribute to the progression of neurodegenerative diseases [207]. The most complete data was collected on the role of neuroinflammation in the pathogenesis of neurodegenerative diseases, and, accordingly, there was the largest number of studies on possible neuroinflammatory markers of AD and MCI in this area.

According to a meta-analysis [208] which analyzed 170 studies, there was an increase in inflammatory blood markers such as IL-1β, IL-6, soluble TNF receptor 1 (sTNFR1), soluble TNF receptor 2 (sTNFR2), alpha1-antichymotrypsin (α1-ACT), C reactive protein, monocyte chemoattractant protein-1 (MCP-1), soluble CD40 ligand, soluble triggering receptor expressed on myeloid cells 2 (sTREM2), CSF levels of IL-10, transforming growth factor-beta 1, chitinase-3-like protein 1 (YKL-40), α1-ACT, NGF and visinin-like protein-1 (VILIP-1) in the AD group compared to healthy controls. Fewer markers were found in the MCI group: sTNFR2, IL-6, MCP-1 and lower levels of IL-8 in the blood, and increased concentrations of VILIP-1, sTREM2 and YKL-40 in CSF. Additionally, increased peripheral sTNFR1 and sTNFR2 levels were observed in AD compared with MCI. Other MCI markers were found as well. Levels of TNF-α and serum amyloid A were higher in participants with MCI compared to cognitively healthy individuals. naMCI was associated with higher levels of TNF-α, serum amyloid A, IL-12 and IL-1β and. Plasminogen activator inhibitor-1 (PAI-1) levels were higher in cognitively normal and naMCI than in aMCI [209]. Numerous studies showed increased levels of pro-inflammatory IL in patients with AD and MCI, for example, in peripheral blood mononuclear cells [210] and cytokines of the cytokines IL-1 family [211].

Cytokines were associated with cognitive impairment in patients [211], and C-reactive protein (CRP) levels were inversely correlated with cognitive and functional decline [212]. IL-4 was found to be positively associated with the volumes of certain brain structures on MRI [213]. There are several studies devoted to the association of scores on various scales of cognitive impairment with markers of neuroinflammation. The MCI group had higher levels of IL-1beta, IL-2, IL-4 and IL-10 compared to the healthy group. Nevertheless, lower levels of IL-1beta and IL-4 were associated with longer duration of memory symptoms. In the prodromal AD group, lower IL-1beta, IL-2, and IL-4 were associated with the increasing duration of memory symptoms [214]. Significant correlations of plasma sTNFR-1 and soluble interleukin-2 receptor subunit alpha (sIL-2Rα) levels with MoCA scores in the whole cohort and the MCI group were obtained by Shen 2019 [215]. Lower Aβ40 and Aβ42 and higher IL-8, IL-10, and TNF-α were associated with greater cognitive decline per the MoCA and Cambridge Cognition Examination (CAMCOG) scores [216]. Higher high sensitivity CRP was significantly associated with poorer basic activities of daily living [217]. A seven-year study analyzed the relationship of markers of neuroinflammation with the progression of cognitive impairment. Higher levels of plasma TNFR1 were associated with a steeper rate of cognitive decline on follow-up and with a larger risk of progression from normal to MCI [207]. IL-6 predicted the progression to MCI. Higher concentrations of sTNFR2 and IL-6 were associated with Aβ burden in those with lower hippocampal volume [218].

The main supplier of cytokines in the CNS is microglia. Activated microglia produces pro-inflammatory cytokines, which can lead to neurodegeneration and depressive disorders through hyperactivation of the hypothalamic-pituitary-adrenal axis and an increase in IDO activity [219]. Microglia activation is indicated by sTREM2, clusterin, and YKL-40. A marker for neuron-microglia communication, CX3CL1, and a marker for microglial mobilization and inflammatory response, MCP-1, are also well-known. The levels of these proteins were altered in patients with neurodegeneration, which indicated the role of microglial activations. The studies [220,221] investigated markers of microglial reaction YKL-40 and sCD14 in CSF. YKL-40 was a glycoprotein produced by inflammatory, cancer, and stem cells. Normally, it was expressed in fibrillar astrocytes in the white matter. YKL-40 was a disease-specific marker of neuroinflammation [222]. The association of this protein with neurodegeneration was shown in many studies. YKL-40 was significantly elevated in AD compared with healthy controls in CSF. This marker differentiated between AD and controls and between stable MCI to AD and those that convert to AD and vascular dementia [221]. The Muszyński (2017) study also showed that CSF concentrations of YKL-40 were significantly higher in MCI and AD patients [223]. Another study demonstrated a significant interaction in the association between YKL-40 levels and gray-matter volume according to ε4 status [224]. YKL-40 could contribute to the reduction of stability and increase the permeability of the BBB in AD patients. Elevated concentrations of YKL-40 correlated significantly with increased albumin quotient and decreased Aβ42/40 ratio in AD patients [225]. Levels of VILIP-1 and YKL-40 in MCI predicted progression to AD in a long-term study [226].

The dysregulation of the complement system also contributed to the neuroinflammatory component of neurodegenerative diseases associated with impaired cognitive function [227]. Of the 53 analytes tested, 10 differed in the control, AD, and MCI groups. Factor H, factor B, MCP-1, soluble complement receptor 1 and eotaxin-1 differentiated AD and control. Soluble complement receptor 1, eotaxin-1 and MCP-1 optimally differentiated AD and MCI. Factor B and factor H plus predicted MCI progression to AD [227]. Lower levels of Complement 3 and Factor H were associated with faster cognitive decline in MCI (AD Assessment Scale-cognitive subscale test (ADAS-Cog)). Lower Factor H levels were associated with larger lateral ventricular volume, which was indicative of brain atrophy [228].

Neutrophil activation also contributed to the progression of neurodegenerative diseases. Thus, the concentrations of myeloperoxidase and neutrophil gelatinase-associated lipocalin were significantly higher in AD relative to control. Peripheral blood concentrations of neutrophil gelatinase-associated lipocalin were also higher in MCI compared to control. However, this relationship was not observed in CSF [229]. Macrophage migration inhibitory factor (MIF) is a pro-inflammatory protein. MIF-related inflammation is related to amyloid pathology, tau hyperphosphorylation, and neuronal injury at the early clinical stages of AD. Its levels were higher in MCI patients with an AD biochemical profile. High MIF levels were associated with higher CSF tau and ptau and lower CSF Aβ1-42, typical AD biomarkers. Higher MIF CSF levels were associated with accelerated cognitive decline in MCI and mild dementia [230]. MCP-1, also known as chemokine CCL2, mediates inflammation in AD. AD patients had higher plasma MCP-1 levels compared with MCI patients and controls, and severe AD patients had the highest levels. MCP-1 level was significantly correlated with changes in the two-year MMSE [231]. Macrophage inhibitory cytokine levels were associated with cognitive performance and cognitive decline [232]. Patients with MCI have stronger Treg-related immunosuppression status compared with patients with probable AD-related dementia [233].

Shen (2019) [215] investigated the number of inflammatory markers and markers of vascular damage as a vascular hypothesis for dementia. In addition to the increased TNF-α and C-peptide, the authors noticed decreased levels of VEGF-A and endothelial plasminogen activator inhibitor 1.

In modern clinical practice, lactate concentration in blood is a measure of anaerobic metabolism used to detect tissue hypoxia. The main reason for excessive lactate formation is an increase in anaerobic processes in tissues with insufficient oxygen supply. Patients with an MCI had an increased concentration of plasma lactate and serum CRP compared to cognitive normal participants. Researchers suggested that concentration of plasma lactate was associated with systemic inflammation and MCI [109]. The role of hypoxia was also shown by [234]. Hypoxia-inducible factor 1 facilitated glycolysis and glucose metabolism and improved cerebral blood flow, which opposed the toxicity of hypoxia. Early adaptation to oxidative and inflammatory stress to attenuate neuronal cell death can be observed through increased expression of proglycolytic factors, cerebral autoregulation mediated by hypoxia-inducible factor 1 (HIF-1) in MCI and in the early AD. Increased HIF-1α expression countered oxidative stress and/or inflammation and was neuroprotective.

Adiponectin is released by the adipose tissue, and regulates energy homeostasis and has anti-atherogenic and anti-inflammatory effects. It also plays a role in the pathophysiology of neurodegenerative disorders. Serum levels of adiponectin were significantly lower in MCI and AD compared to controls. Lower adiponectin levels were associated with cognitive dysfunction [235].

Inflammatory processes secondary to neuroinflammation can cause telomere depletion. Analysis of leukocyte telomere length showed a progressive decrease in the series control > MCI > patients with AD. A general decrease in telomere length correlated with increased plasma IL-1β levels [236]. S100A9 is a pro-inflammatory protein. Its level in CSF correlated with Aβ (1-42) levels. Its concentration decreased at the stable MCI stage and then progressively decreased in MCI due to AD (MCI-AD), AD, and vascular dementia patients [237].

Moreno-Rodriguez (2020) [238] performed an immunochemical analysis of the frontal cortex and white matter of patients diagnosed with MCI and AD during their lifetimes. Chitinase 3-like 1 (CHI3L1) and chitinase 3-like 2 (CHI3L2) were inflammatory markers of AD. CHI3L1-immunoreactive astrocyte numbers were increased in the frontal cortex and white matter in AD compared to controls. Glial fibrillary acidic protein and ionized calcium-binding adapter molecule 1 immunoreactive cell numbers were increased in MCI compared to control in white matter. Iba1, CD44 with CHI3L2, and GFAP protein levels were associated with disease progression. CHI3L1 and Iba1 cell numbers in white matter demonstrated a significant association with episodic memory and perceptual speed [238].

An indicator of inflammation involvement in the pathogenesis of neurodegeneration is that therapeutic agents with anti-inflammatory effects improve the level of cognitive impairment indicators. Homotaurine is a potential therapeutic compound for AD, as it has anti-inflammatory properties. After taking homotaurine, patients carrying the APOEε4 allele demonstrated a significant decrease in the level of IL-18 (but not other cytokines) [239].

Multiple proteins were associated with baseline cortical thickness and cognitive performance in early and later-stage AD: ROBO2, ischemic injury proteins (uPA, SMOC2, tPA), mTOR and Wnt/β-catenin signaling proteins (AXIN1, EIF4EBP1), and glucose metabolism protein HAGH [240]. Methylglyoxal (MG) and glyoxal (GO) could also be candidates for biomarkers, as their blood levels were higher in the MCI group compared to the control. The levels of MG in the serum are sensitive enough to differentiate MCI from controls but not from AD. Meanwhile, serum GO levels allow for the differentiating of MCI from control and AD groups [241].

Lower plasma fatty acids (eicosapentaenoic acid (EPA) and DHA) levels and higher reptin and low-density lipoprotein levels were associated with AD [242].

Changes in various metabolic pathways might be involved in the pathogenesis of neurodegenerative diseases. A large meta-analysis of publications since 1984 identified potential biomarkers such as insulin growth factor binding protein-2 (IGFBP-2), pancreatic polypeptide (PP), alpha-2-macroglobulin (α2M), apolipoprotein A-1 (ApoA-1), and fibrinogen-γ-chain identified. Those findings provided important insights into AD blood biomarkers discovery via proteomics [243].

The dysregulation of NMDAR is a part of AD pathophysiology. D-amino acid oxidase (DAO) and amino acids can regulate the NMDAR receptor function. The DAO levels increase with the severity of the cognitive deficits, according to CDR assessment results. DAO levels were significantly associated with D-glutamate and D-serine levels [244]. NMDAR antibodies could play a role in neurodegenerative disorders. Antibodies to NMDAR were detected in groups of patients with AD, ischemic vascular dementia, frontotemporal dementia, patients with Lewy body disease, patients with MCI, and even in aged but healthy volunteers without neuropsychiatric disorders. The presence of NMDAR antibodies in dementia could influence the incidence of comorbid depressive and/or psychotic states [245]. NMDAR antibodies were present in different types of dementia and elderly healthy individuals. In combination with disturbed B-CSF-B integrity, they seemed to promote their pathological potential on cognitive decline [246]. Also, plasma homocysteic acid (HCA) could be a useful indicator as an early diagnostic marker for MCI. HCA exhibited very high brain toxicity at low concentrations and overactivated the NMDAR. HCA seemed to be upstream from neurodegeneration in the AD pathology because it was known that an overactive NMDAR promoted amyloid polymerization and tau phosphorylation in AD [247].

Analysis of 270 proteins in CSF and plasma revealed different regulation of their innate and adaptive immunity, membrane phospholipids, axon guidance, cell adhesion and differentiation, energy metabolism, and amino acids and lipids pathways that might be involved in early AD development [248]. Disruption of tryptophan metabolism was also observed in AD. Researchers noticed lower metabolite concentrations of tryptophan pathway metabolites in the AD group: serotonin, 5-hydroxyindoleacetic, kynurenine, tryptophan, and xanthurenic acid. For each listed metabolite, a decreasing trend in concentrations was observed that was in line with clinical diagnosis, the worst being with AD, while MCI had intermediate results [249,250].

The greatest role of vascular disorders is in the formation of neurodegenerative diseases. Neurodegeneration, the pathogenesis of which is dominated by a vascular component, is called vascular dementia and is associated with atherosclerosis of the cerebral vessels. Traditional cardiovascular risk factors such as obesity, high cholesterol, stress, diabetes, etc., are also factors in neurodegenerative diseases and the development of dementia [66]. Cerebral small vascular disease is thought to be responsible for about 50% of all dementias worldwide, including AD [251].

Vascular cognitive impairment manifests in deficits in attention and executive functioning. This is in contrast to episodic memory deficits, which are more prominent in Alzheimer’s dementia [252]. Neuroimaging techniques show impaired brain perfusion in different types of dementia.

However, with regard to circulating markers in the blood, it is important to select those that can specifically indicate cognitive impairment and predict cognitive decline. Such markers in the blood may be important for early diagnosis of cognitive decline in vascular risk factors. These are markers of inflammation, endothelial dysfunction, and vascular thrombosis. Inflammatory markers in neurodegeneration have been discussed above.

Miralbell (2013) [253] examined asymmetric dimethylarginine (ADMA) as a marker of endothelial dysfunction and the plasminogen activator inhibitor 1 (PAI-1) as a marker of vascular thrombosis. Increasing levels of ADMA have been described in AD and explained lower performance in verbal memory [254]. Elevated levels of PAI-1 have been found in naMCI and vascular dementia patients [253].

Vascular changes in AD have been typically attributed to the vasculotoxic effects of Aβ [251]. Disruption of the blood–encephalic barrier has been observed in the early stages of cognitive decline. A paper [251] examined the BBB-associated capillary mural cell pericyte marker, a soluble platelet-derived growth factor receptor-β (sPDGFRβ)8. The authors found an increase in sPDGFRβ in CSF with more progressive CDR impairment. These same authors believe that vascular damage in dementia is not related to the vasculotoxic effects of Aβ, as previously thought [251].

Levels of VEGF, intercellular adhesion molecule-1 (ICAM-1), and vascular cell adhesion molecule-1 (VCAM-1) are associated with aging and neurodegeneration [119]. VEGF is an important factor in preventing cognitive decline in neuropsychiatric diseases [115]. ICAM-1 and VCAM-1 are markers of endothelial cell activation. Cerebral hypoperfusion increases the expression of ICAM-1 and VCAM-1 and promotes oxidative stress, which leads to peroxynitrite formation, lipid peroxidation, matrix metalloproteinases (MMPs) activation, and DNA damage [255]. Accordingly, these markers of endothelial dysfunction may be markers of cognitive impairment in neurodegenerative diseases.

Coagulation and fibrinolytic pathways are involved in cerebrovascular disease. However, little is known about the role of hemostatic biomarkers. High levels of fibrinogen, Lp PLA2 (an enzyme that affects the degradation of platelet-activating factors to inactive), high plasma levels of von Willebrand factor, homocysteine, and others have been shown to be associated with dementia [255].

Aβ can cause mitochondrial dysfunction in AD, and Aβ is found in mitochondria in the brains of AD patients [72]. However, there is no unequivocal opinion in the literature as to whether mitochondrial dysfunction is a cause or consequence of the effect of Aβ aggregation. The authors suggest that people with a primary impaired energy metabolism, a typical AD phenotype, but without plaques, could potentially represent “plaque-negative” cases of AD. The role of tau proteins in mitochondrial dysfunction has also been shown [256]. In genetic animal models of AD, bioenergetic mitochondrial deficiency has been shown to occur at a very early age, and mitochondrial dysfunction precedes the pathology of AD [73]. In addition, mitochondria serve as a link between lipid metabolism, glucose metabolism, inflammation, oxidative stress and AD pathogenesis [73]. ROS and RNS play a central role in aging. The role of oxidative stress in aging is supported by the following facts: (a) species longevity is related to antioxidant activity; (b) enhanced expression of antioxidant enzymes increases longevity; (c) free radical damage and protein nitrosylation increase with age; (d) reduced caloric intake reduces the production of reactive forms and increases longevity [156]. The same processes have been shown to be characteristic of the pathogenesis of neurodegenerative diseases [156]. Mitochondrial deficits were observed in early AD, so this opens up the prospect of finding biomarkers associated with AD and emerging cognitive dysfunction [256]. However, metabolic markers that can be detected in blood or CSF have not been identified at this time. Studies in transgenic mouse models of AD have shown early disturbances in the following mitochondria-related metabolic pathways: the citric acid cycle, oxidative phosphorylation system, pyruvate metabolism, glycolysis, oxidative stress, fatty acid oxidation, ketone body metabolism, ion transport, apoptosis, and mitochondrial protein synthesis. The number of mitochondrial proteins, including complex I and complex IV subunits, is reduced in the cerebral cortex in AD [257]. At the same time, other mitochondrial proteins, including malate dehydrogenase and succinate dehydrogenase, show increased activity in the cerebral cortex in AD [258]. Measuring the NAD+/NADH ratio or Complex II levels can be a useful marker of mitochondrial spare respiratory capacity (MSRC), as Complex II activity has been shown to be directly related to MSRC [259]. Complex IV expression can also be evaluated, but it would not be a clear measurement of mitochondrial function [256]. Impaired mitochondrial enzyme function has also been shown [73]. A metabolomic assay may be useful to look for markers. Krebs cycle markers in the CSF and liver were significantly reduced in patients with MCI compared to those without the disease [73]. Several studies have shown that lipid oxidation and superoxide dismutase activity were increased, whereas complex IV, complex III, mitochondrial membrane potential, and ATP were decreased in platelets from patients with AD, and elevated levels of oxidative DNA damage, Mn-superoxide dismutase (SOD2) mRNA, 4-HNE, and 3-nitrotyrosine were observed in lymphocytes from AD patients [73]. The identification of sensitive, reliable, and easily detectable mitochondrial biomarkers is still a major challenge, despite advances in understanding the abnormalities in the biochemical reaction cascades that occur in mitochondria in pathology.

Thus, many biomarkers of cognitive decline have been found, mainly in pre-existing Alzheimer’s disease. However, the most valuable biomarkers are those that would predict the progression of minor cognitive decline into dementia for early diagnosis and treatment of pathology. Despite many studies, the representative marker with a high predictive rate for dementia has not been found yet. Apparently, this should be not a single marker but a whole panel that would reflect all aspects of the cognitive impairments pathogenesis.

Summary of possible candidate biomarkers of cognitive impairments in mental disorders and neurodegenerative diseases are in Table 1.

## 5. Conclusions and Future Directions

In this review, we have considered the important components of the pathogenesis of cognitive impairment in mental illness. The involvement of genetic and epigenetic mechanisms in the formation of cognitive disorders results in metabolic and proteomic alterations. The mechanisms underpinning cognitive dysfunction in psychiatric disorders are complex. Nonetheless, accumulating evidence supports a tiered mechanism involving gene polymorphisms, epigenetic regulation of DNA, and subsequent changes in the proteome and metabolome as common features underlying cognitive disabilities. Despite the different clinical diagnoses in which the cognitive sphere is impaired, one cannot fail to note some commonality in the changes observed at both the genetic and metabolic levels.

Plasma biomarkers, including neurotrophic factors, pro-inflammatory cytokines, and markers of oxidative stress are persistently increased in a large proportion of patients with cognitive dysfunction and, thus, can determine symptoms.

In addition, these plasma markers depend, of course, on the DNA sequences of genes and transcription mechanisms. Moreover, external factors such as lifestyle, diet, bad habits, and stress levels regulate gene transcription through epigenetic mechanisms of methylation and acetylation of DNA and histones. The genetic basis of psychiatric disorders is complex and still unclear.

Data on the neuroinflammatory processes that are implicated in cognitive-associated psychiatric disorders and how they promote the development, progression, and maintenance of the disorders are limited.

In our review, we didn’t touched the neurophysiological, fMRI and PET markers; we focused on circulating molecular biological markers, which are associated with cognitive decline.

In clinical medicine, the biomarkers are called symptoms, but in biochemistry, there are other molecular biomarkers, that is, molecules that can be measured in different ways. These molecules can range from simple ions to high molecular weight proteins, such as antibodies to various antigens.

Clinical psychiatry is in dire need of developing and using biomarkers. We are referring to prognostic and predictive biomarkers that are valuable before and after diagnosis. Without such markers, it is difficult to imagine personalized medicine, which implies the division of patients with the same diagnosis into groups and a rational choice of different types of prevention and treatment for them. The biomarker panel for each case should be multimodal and provide information on the functional state of the CNS at all levels, from DNA structure and epigenetic regulation to quantitative content of neurotrophic, inflammatory, vascular, and oxidative factors. The more physicians are aware of the biological changes in the patient’s body, the more likely it is that the most effective treatment with the fewest side effects will be chosen. The introduction into clinical practice of mandatory standardized tests assessing the quantification of plasma biomarkers reflecting neuroinflammation, neurotrophic function, and oxidative stress levels, together with knowledge of genetic predisposition to disease and cognitive decline, should be made routine, along with clinical interview and psychometric techniques.

To do so, the research community needs to focus on selecting biomarkers with the best diagnostic accuracy, sensitivity, and specificity from the many possible ones. Without validated analytical multimodal panels, the transition to personalized psychiatry is impossible.

## Figures and Tables

**Table 1 ijms-23-01217-t001:** Summary of possible candidate biomarkers of cognitive impairments in mental disorders and neurodegenerative diseases. AD—Alzheimer’s disease, APOE4—the epsilon 4 allele of the apolipoprotein gene, BBB—blood brain barrier, BDNF—brain-derived neurotrophic factor, CNS—central nervous system, CRP—C-reactive protein, CSF—cerebrospinal fluid, GWAS—genome-wide association study, IL—interleukin, MCI—mild cognitive impairment, ROS—reactive oxygen species, SZ—schizophrenia, SNP—single nucleotide polymorphism, TNF-α—tumour necrosis factor-α.

Biomarker	Biomarker Group	Bodily Fluid	Effect in Organism	Change in Disease	Reference
Schizophrenia					
Transcription Factor 20 (TCF20)	Genetic	-	TCF20 encodes a widely expressed transcriptional coregulatory.	TCF20 mutations are associated with autism and intellectual disability. A shared genetic effects between SZ and cognitive traits.	[75]
Cytochrome P450 Family 2 Subfamily D Member 6 (CYP2D6)	Genetic	-	CYP2D6 encodes a cytochrome P450 enzyme that metabolizes a broad range of drugs, including antipsychotics, and may also be involved in the metabolism of neurotransmitters, including serotonin and dopamine.	A shared genetic effects between SZ and cognitive traits.	[75]
Alpha-N-Acetylgalactosaminidase (NAGA)	Genetic	-	NAGA encodes a lysosomal enzyme that modifies glycoconjugates.	A shared genetic effects between SZ and cognitive traits.	[75]
Protein interacting with C kinase 1 alpha (PICK1)	Genetic	-	PICK1 colocalizes with AMPA receptors at excitatory synapses, induces AMPA receptor synaptic activity with its internalization and down-regulation. Phosphorylation of AMPA receptors by PKCα bound to PICK1 causes the activation of NMDA receptor.	PICK1 polymorphisms may associate with cognitive functions in SZ patients. The rs2076369 G/T genotype showed better performance than T/T homozygotes, and A/A homozygotes of rs3952 performed better than G/G in the cognitive scores.	[75]
Encoding chromodomain helicase DNA binding protein 7 (CHD7)	Genetic	-	ATP-dependent chromatin remodeler.	Strongest associations SNP rs6984242 with IQ and episodic memory in SZ.	[77]
E1A Binding Protein P300 (EP300)	Genetic	-	EP300 encodes the histone acetyltransferase.	The associations SNP rs9607782 with episodic memory in SZ.	[77]
GATA Zinc Finger Domain Containing 2A (GATAD2A)	Genetic	-	A transcriptional repressor and subunit of the nucleosome remodeling and deacetylase (NuRD) complex.	The associations SNP rs2905426 with IQ in SZ.	[77]
Lysine Demethylase 3B (KDM3B)	Genetic	-	Gene necessary for normal spermatogenesis and sexual behaviors in males and encodes an enzyme that removes a key transcriptional repressive modification from chromatin.	The associations SNP rs10043984 with attention in SZ.	[77]
Arginine-Glutamic Acid Dipeptide Repeats (RERE)	Genetic	-	A transcriptional co-repressor that binds chromatin and is involved in cerebellar development.	The associations SNP rs34269918 with attention in SZ.	[77]
Solute Carrier Family 1 Member 2 (SLC1A2)	Genetic	-	Gene encodes a member of a family of solute transporter proteins, which clears the excitatory neurotransmitter glutamate from the extracellular space at synapses.	SNP rs4354668 and its haplotypes may be involved in impaired executive function in SZ.	[260]
Catechol-O-methyltransferase (COMT)	Genetic	-	An enzyme that plays an important role in the degrade of catecholamines.	The associations between the COMT genotype (COMT Val(158)Met) and indices of attention/executive functions in patients with SZ.	[261,262]
Cell-free mitochondrial DNA (cf-mtDNA)	Genetic	Blood	Circulating cf-mtDNA fragments in blood plasma.	Circulating cf-mtDNA levels may serve as a potential biomarker to determine the cognitive status of patients with SZ.	[121]
Immunoglobulin GM (γ marker) and KM (κ marker)	Genetic Inflammatory	-	GM and KM allotypes—genetic markers of immunoglobulin γ and κ chains, which are associated with humoral immunity.	Particular KM and GM genotypes associate with verbal memory and attention and processing speed scores. Epistatic effects of GM and KM genotypes on attention and processing speed, verbal fluency, and motor speed.	[134,263]
IL-10	Genetic Inflammatory	-	The anti-inflammatory cytokine.	SZ patients with the AA allele of the IL10-592 A/C polymorphism perform worse in attention.	[78]
IL-1β mRNA	Epigenetic Inflammatory	Blood	The anti-inflammatory cytokine.	The elevated IL-1β mRNA levels are associated with both impairments in verbal fluency and brain volume reduction in patients with SZ.	[104]
Histone deacetylases (HDAC)	Epigenetic	-	HDACs modify histones and change chromatin conformation and play an important role in the regulation of gene expression. HDAC regulate cognitive circuitry.	Relative HDAC expression is lower in the dorsolateral PFC of patients with SZ compared with controls, and HDAC expression positively correlated with cognitive performance scores.	[79]
BDNF	Neuroplasticity	Blood	BDNF is a neurotrophin in the brain, whose functions are to control neuronal and glial development, neuroprotection, and modulation of synaptic interactions.	Cognitive impairment in SZ is associated with decreased levels of BDNF.	[93,263,264,265]
Insulin-like growth factors—1 (IGF-1)	Neuroplasticity	Blood	IGFs are members of the insulin superfamily and play a key regulatory role in the development of the brain. IGFs promote the proliferation, differentiation, and maturation of neural cells.	IGF-1 levels correlate positively with executive function and attention scores in SZ patients. IGF-1 is an independent contributor to deficits in executive function and attention among SZ patients.	[266]
Neuron-specific enolase (NSE)	Neuroplasticity	Blood	NSE is the biomarker of all differentiated neurons. The determination of the concentration of NSE in the serum and CSF provides information on the severity of neuronal damage and the integrity of the BBB.	Thought disorders are more pronounced in patients with higher NSE levels.	[267]
CRP	Inflammatory	Blood	Acute inflammation protein that is overexpressed in inflammatory conditions.	Cognitive impairment in SZ is associated with elevated levels of CRP. Thought disorders are more pronounced in patients with higher CRP levels.	[93,102,103,106,267,268]
TNF-α	Inflammatory	Blood	Pro-inflammatory cytokine.	A better cognitive functioning of SZ patients with higher levels of TNF-α.	[102,103]
IL-6	Inflammatory	Blood	The anti-inflammatory cytokine.	A positive association between IL-6 levels and worse cognitive performance.	[103,105,106]
IL-1 receptor antagonist (IL-1Ra)	Inflammatory	Blood	IL-1RA is a protein that regulates the activity of IL-1.	General cognitive abilities is associated with IL-1Ra in SZ patients.	[103]
Matrix metalloprotease-9 (MMP-9)	Inflammatory	Blood	MMPs are secreted by glial and neuronal cells in the brain and are involved in neuroinflammation and neurotoxicity, which play a role in hippocampal-dependent learning.	MMP-9 is associated with fluency and language component of cognition and increases the risk of cognitive impairment in SZ.	[269]
Asymmetric dimethylarginine (ADMA)	Other	Blood	Endogenous inhibitor of the nitric oxide synthase	ADMA associate negatively with attention, working memory and executive function in SZ.	[181]
Hydrogen sulfide (H_2_S)	Other	Blood	H_2_S is an endogenous gasotransmitter, that regulates NMDAR function.	A positive association between H_2_S levels and working memory, visual memory, or executive function in SZ patients.	[181]
3-methoxy-4-hydroxyphenylglycol (MHPG)	Other	Blood	Plasma catecholamine metabolite (norepinephrine degradation).	MHPG levels associate with working memory, verbal fluency, executive function, attention, and processing speed in SZ	[263,270]
Metabolic syndrome (MetS)	Other	-	MetS is defined as a clustering of at least three interrelated cardiovascular risk-factor abnormalities, including abdominal obesity, hyperglycemia, hypertension, high triglycerides, or low high-density lipoprotein (HDL) cholesterol levels.	A relationship between each of the components of MetS and cognitive impairment in SZ.	[271]
**Depression**					
Solute Carrier Family 27 Member 1 (SLC27A1)	Genetic	-	SLC27A1 encodes the fatty acid transport protein 1, which has DHA as a substrate. DHA is an endogenous neuroprotective compound in the brain.	Each additional copy of the G allele of SNP rs11666579 is associated with an average decrease of baseline cognitive scores (CERAD-TS) in late-life depression patients.	[129]
Glutaredoxin And Cysteine Rich Domain Containing 1 (GRXCR1)	Genetic	-	A gene previously linked with deafness.	SNP rs73240021 is the most significant SNP associated with cognitive scores (CERAD-TS) decline over time in late-life depression patients.	[129]
SNP rs1766259	Genetic	-	Intergenic SNP on chromosome 6.	For each additional G allele of SNP rs17662598, average baseline cognitive scores (CERAD-TS) decrease in late-life depression patients.	[129]
STin2 polymorphism	Genetic	-	The STin2 polymorphism is a tandem repeat located in intron 2 in the serotonin transporter gene. STin2 polymorphism has an effect on the quantity of the serotonin transporter.	The frequencies of STin2 genotypes differ in depressed and controlled patients. STin2 genotypes influence on results of cognitive interference tasks, working memory tasks, and recall tasks in depressed patients.	[123]
Neuronal thread protein (AD7c-NTP)	Neuroplasticity	Urine	A transmembran phosphoprotein that causes apoptosis and neuritic sprouting in transfected neuronal cells.	Urinary levels of AD7c-NTP in the late-life depression with cognitive impairment patients are higher than in both the late-life depression without cognitive impairment patients, and healthy control, but lower than in the AD patients.	[143,272]
CRP	Inflammatory	Blood	Acute inflammation protein that is overexpressed in inflammatory conditions.	Persistent depressive symptoms (including cognitive) are associated with subsequent higher levels of CRP. Among women, higher CRP is associated with increased severity of cognitive symptoms. Baseline CRP predicts cognitive symptoms of depression at follow-up. The highest CRP quintile is associated with both negative and positive differences in cognitive performance. The lower CRP levels are associated with improved performance in psychomotor speed tasks.	[147,148,149,273,274]
Interferon gamma (IFN-γ)	Inflammatory	Blood	Cytokine	IFN-γ negatively correlated with the score cognitive factor.	[150]
TNFα	Inflammatory	Blood	Pro-inflammatory cytokine.	TNF-α negatively correlated with the score cognitive factor.	[150]
IL-1β	Inflammatory	Blood	Cytokine	Partial changes in cognitive function and changes in IL-1β are correlated in treatment-resistant depression patients.	[151]
IL-4	Inflammatory	Blood	Cytokine	IL-4 negatively correlated with the score cognitive factor.	[150]
IL-5	Inflammatory	Blood	Cytokine	IL-5 negatively correlated with the score cognitive factor.	[150]
IL-6	Inflammatory	Blood	Cytokine	Baseline IL-6 predicted cognitive symptoms of depression at follow-up. The lower IL-6 levels are associated with improved performance of the Stroop, incongruent test. Poorer verbal fluency performance is associated with reduced IL-6 levels.	[147,149]
IL-12	Inflammatory	Blood	Cytokine	IL-12 negatively correlated with the score cognitive factor.	[150]
IL-13	Inflammatory	Blood	Cytokine	IL-13 negatively correlated with the score cognitive factor.	[150]
**MCI/Alzheimer’s disease**					
Aβ40, Aβ42, Aβ42/Aβ40, Aβ42/Aβ38 ratios	Neurodegenerative	CSF	Amyloid-beta (Aβ)—peptides that are the main component of amyloid plaques in the brain of patients with neurodegenerative diseases. The Aβ42 is a major component of senile plaques and contributes to cerebral amyloid angiopathy in AD.	The lower CSF Aβ42 concentrations indicating higher levels of brain Aβ42. Reduced levels of Aβ42 can be detected in MCI, in the pre-clinical stages of AD, and in AD. The Aβ level is lower in MCI than in AD. The CSF Aβ42/Aβ40 and Aβ42/Aβ38 ratios are better than CSF Aβ42 to detect brain amyloid deposition in prodromal AD and to differentiate AD dementia from non-AD dementias. Aβ40 levels are higher in AD patients.	[39,162,273,274,275,276,277,278]
Aβ40, Aβ42 and Aβ40/Aβ42 ratio		Blood		Low plasma Aβ levels, including Aβ40, Aβ42, and Aβ42/Aβ40 ratio may indicate a cognitive decline in MCI. Lower Aβ40 and Aβ42 are associated with greater cognitive decline in individuals with cognitive deterioration and/or progression to MCI/probably AD.	[216,275,279]
Aβ42		-		The brain amyloid accumulation in MCI and AD (neuroimaging).	[39,276]
Total tau (T-tau) and phosphorylated tau (P-tau)	Neurodegenerative	CSF	The normal function of tau protein is to bind to and stabilise microtubules in neuronal axons, a process that is inhibited when tau becomes phosphorylated. Abnormally phosphorylated and truncated tau proteins are the major component of neurofibrillary tangles in AD.	The elevated CSF concentrations of T-tau and P-tau indicate a neuronal injury and predict progression from MCI to AD-related dementia. The tau level is lower in MCI than in AD. However, increased levels of CSF T-tau are not specific to AD.	[39,273,274,275,276,280]
		Blood		In AD, plasma T-tau levels are increased, but less so than in the CSF, and there is no detectable increase in the MCI stage of the disease.	[162,276,280]
APOE4	Genetic	-	Apolipoproteins such as ApoE bind to low-density lipoprotein receptors thereby mediating transport of cholesterol and other lipoproteins. ApoE is encoded by the APOE gene, located on chromosome 19, which has three major alleles: ε2, ε3 and ε4.	APOE4 confers risk for MCI, as it does for AD. The 1 or 2 ε4 alleles in the APOE gene increases confidence in the diagnosis of MCI due to AD.	[39,281,282]
	Epigenetic	Blood		Elevated plasma ApoE and APOE methylation of CpGs 165, 190, and 198 are risk factors for MCI. Higher CpG-227 methylation correlates with a lower risk for MCI. CpG-227 is correlated with plasma ApoE levels.	[186]
Amyloid Beta Precursor Protein (APP)	Neurodegenerative Genetic	-	A transmembrane neuronal protein. Sequential cleavage of APP by β-secretase and then by γ-secretase produces Aβ peptides.	The APP mutations affect a common pathogenic pathway in APP synthesis and proteolysis, which leads to excessive production of Aβ.	[167,283,284]
Beta-Secretase 1 (BACE1)	Genetic Neurodegenerative	-	BACE1 gene encodes a member of the peptidase A1 family of aspartic proteases. BACE1 catalyzes the initial cleavage of the APP to generate Aβ.	GG genotype and G allele of SNP rs638405 probably increase the risk of AD. SNP rs638405 decreased the risk of APOE4 positive AD patients.	[75,189]
	Epigenetic Neurodegenerative	Blood		BACE1 mRNA levels are increased in peripheral blood mononuclear cells from AD patients along with an increase in promoter accessibility and histone H3 acetylation.	[97,285]
	Neurodegenerative	CSF		Patients with AD and MCI have higher CSF BACE1.	[286,287]
		Blood		The increased BACE1 activity in serum may represent a potential biomarker for late-onset AD.	[288]
Presenilin 1 (*PSEN1*), Presenilin 2 (*PSEN2*)	Neurodegenerative Genetic	-	PSEN1 and PSEN2 are critical components of the γ-secretase complex (APP cleavage)	Gene associated with AD risk.	[167,168,169,170,171,172,173,174,175,176,177,178,179,180,181,182,183,184,185,186,187,188,189,190,191,192,193,194,195,196,197,198,199,200,201,202,203,204,205,206,207,208,209,210,211,212,213,214,215,216,217,218,219,220,221,222,223,224,225,226,227,228,229,230,231,232,233,234,235,236,237,238,239,240,241,242,243,244,245,246,247,248,249,250,251,252,253,254,255,256,257,258,259,260,261,262,263,264,265,266,267,268,269,270,271,272,273,274,275,276,277,278,279,280,281,282,283,284,285,286,287,288,289,290,291]
Sortilin related receptor L (SORL1)	Genetic	-	SORL1 is involved in vesicle trafficking from the cell surface to the Golgi-endoplasmic reticulum. SORL1 encodes a key protein involved in the processing of the APP and the secretion of the Aβ peptide.	Gene associated with AD risk. SORL1 expression has been reduced in the brain of MCI patients and may affect the severity of the disease.	[167,292,293,294]
	Epigenetic	Blood		DNA methylation of the SORL1 5’-flanking region may influence the manifestation of MCI in type 2 diabetes mellitus, and might be associated with its neurocognitive presentation.	[189]
Visinin-like protein 1 (VILIP-1)	Neurodegenerative	CSF	VILIP-1 influences the intracellular neuronal signaling pathways involved in synaptic plasticity.	VILIP-1 is involved in calcium-mediated neuronal injury, which leads to increased levels of VILIP-1 in CSF. VILIP-1 can be used as a predictor of cognitive decline in the early stages of AD. Levels of VILIP-1 in MCI predicted progression.	[208,226,292,295,296]
Clusterin (CLU)	Genetic	-	CLU is a stress-activated chaperone protein (apolipoprotein J) that functions in apoptosis, complement regulation, lipid transport, membrane protection, and cell-cell interactions. CLU likely influences Aβ clearance, amyloid deposition, and neuritic toxicity.	CLU genetic variants can affect cognitive function by altering amyloid and lipid (cholesterol) metabolism. CLU gene variants can affect plasma clusterin levels and possibly predict the progression of MCI in AD.	[167,292,297,298,299]
	Epigenetic	-		The expression of CLU is clearly increased after neuronal injuries and degeneration as well as during aging and neurodegenerative diseases.	[299]
	Other	CSF		CLU levels are increased in the brain and CSF of patients with AD.	[299]
		Blood		Plasma CLU levels may change during neurodegeneration. CLU is a risk factor for AD. Patients with MCI with higher CLU levels may have a higher risk of AD progression. CLU plasma levels are positively associated with brain atrophy, disease severity, and disease progression.	[167,292,299,300]
Complement receptor 1 (CR1)	Genetic Inflammatory	-	CR1 is a component of the complement response. CR1 expression on phagocytic cells, such as erythrocytes, results in the ingestion and removal of complement activated particles.	Gene associated with AD risk. Expression of complement factors are reportedly upregulated in affected regions of AD brains. The elevated complement cascade activity could exacerbate AD pathology.	[167]
	Inflammatory	Blood		Soluble CR1 optimally differentiated AD and elderly control, AD and MCI.	[227]
Toll-like receptor 4 (TLR4)	Genetic Inflammatory	-	TLRs belong to the family of pathogen-sensing receptors and contribute to innate immune defense against infection. TLR4 signaling plays a role in amyloid peptide clearance and protects nerve cells against neurodegeneration. TLR4 promotes binding of fibrillar amyloid and its phagocytosis by microglia in AD.	TLR4 can affect the early stages of neurodegeneration and MCI through disruption of microglia activation. The minor allele of the SNP rs4986790 (G) is associated with a reduced risk of developing AD and higher visuospatial and constructional abilities.	[266,292,301,302,303]
Triggering Receptor Expressed On Myeloid Cells 2 (TREM2)	Genetic	-	TREM2 is a receptor expressed on microglia that stimulates phagocytosis and suppresses inflammation. TREM2 may play an important role in neurodegeneration, possibly in clearance of protein aggregates or in neuroinflammatory mechanisms.	Gene associated with AD risk.	[167,294,301]
	Inflammatory	CSF		The increased levels of soluble TREM2 in AD patients.	[208]
Erythropoietin-Producing Hepatoma Receptor A1 (EPHA1)	Genetic	-	EPHA1 is a member of the ephrins family of tyrosine kinase receptors, which plays roles in cell and axonal guidance and synaptic plasticity.	Gene associated with AD risk.	[167]
Bridging integrator 1 (BIN1)	Genetic	-	BIN1 is involved in regulating endocytosis and trafficking, immune response, calcium homeostasis and apoptosis.	Gene associated with AD risk.	[167]
Adaptor Related Protein Complex 2 Subunit Alpha 2 (AP2A2)	Genetic	-	A subunit of the AP-2 adaptor protein complex, which is involved in linking lipid and protein membrane components with the clathrin lattice.	AP2A2 is associated with inferior language function (repetition) in probable amnestic MCI patients.	[47]
Heparan Sulfate-Glucosamine 3-Sulfotransferase 1 (HS3ST1)	Genetic	-	A key component in generating a myriad of distinct heparan sulfate fine structures that carry out multiple biologic activities.	The association between HS3ST1 and working memory for the amnestic MCI patients.	[47]
ATP-binding cassette transporter A7 (ABCA7)	Genetic	-	A member of the ABC transporter superfamily, where it functions to transport substrates across cell membranes.	Gene associated with AD risk. ABCA7 may influence AD risk via cholesterol transfer to ApoE or by clearing Aβ aggregates.	[167,294]
Angiotensin converting esterase (ACE)	Genetic	-	ACE gene encodes an enzyme involved in blood pressure regulation and electrolyte balance.	The D-allele in ACE may serve as potential risk factor for MCI.	[75,292]
		Blood		The elevated ACE levels in serum may serve as potential risk factor for MCI.	
ESR1 and ESR2	Genetic	-	Estrogen receptors genes.	The combination of some genetic variants of ESR genes with APOE4 may increase the risk of amnestic MCI and AD, especially in women.	[292,302,303]
LDL Receptor Related Protein 6 (LRP6)	Genetic	-	LRP6 is a coreceptor in WNT signaling and plays an important role in brain function by supporting synaptic structure and function.	LRP6 gene deficiency can cause memory impairment by affecting learning and memory. LRP6 may be involved in the onset of neurodegeneration due to dysfunctions of long-term potential and immune activation. Abnormal LRP6 can also lead to amyloid production and aggregation.	[143,292]
Translocase Of Outer Mitochondrial Membrane 40 (TOMM40)	Genetic	-	Tom40 protein is a translocase of the outer mitochondrial membrane, which is adjacent to and in linkage disequilibrium with APOE.	TOMM40 polymorphisms have been described as a risk factor for the progression of MCI-AD. TOMM40 may also affect age-related memory functions.	[47,292,304]
CD33	Genetic	-	CD33 protein is a member of the sialic acid-binding Ig-like lectin family of receptors and is expressed on myeloid cells and microglia.	Gene associated with AD risk. CD33 may play an important role in Aβ clearance and other neuroinflammatory pathways mediated by microglia in the brain.	[167]
MS4A locus	Genetic	-	MS4A is a locus that contains several genes associated with the inflammatory response.	Genes associated with AD risk.	[167]
Phospholipase D3 (PLD3)	Genetic	-	A member of the Phospholipase D protein family, that catalyze the hydrolysis of phophatidylcholine to generate phosphatidic acid.	Gene associated with AD risk. Over-expression of PLD3 leads to a decrease in intracellular APP and extracellular Aβ42 and Aβ40, while knock-down of PLD3 leads to a increase in extracellular Aβ42 and Aβ40.	[167,305]
Phosphatidylinositol binding clathrin assembly protein (PICALM)	Genetic	-	A protein involved in clathrin assembly.	Gene associated with AD risk. PICALM-mediated Aβ generation and clearance may influence accumulation of Aβ in AD brains.	[167]
CD2 associated protein (CD2AP)	Genetic	-	A scaffolding protein that is involved in cytoskeletal reorganization and intracellular trafficking.	Gene associated with AD risk.	[167]
Benzodiazepine-associated protein 1 (BZRAP1-AS1, TSPOAP1)	Genetic	-	A subunit of the benzodiazepine receptor complex in mitochondria and a marker of neuroinflammation.	GWAS AD with SNPs in *BZRAP1-AS1.*	[306]
Solute carrier family 24 member 4 (SLC24A4)	Genetic	-	This gene encodes a member of the potassium-dependent sodium/calcium exchanger protein family (NCKX), that is bidirectional membrane transporters.	SLC24A4 cis-SNPs associate with late-onset AD risk. SNP rs10498633 is revealed to be closely related to the risk of late-onset AD in a large GWAS.	[181,307]
Methionine sulfoxide reductase B3 (MSRB3)	Genetic	-	ROS oxidize protein methionine residues. The resulting methionine sulfoxides can be repaired by reductases such as MSRB3.	The MSRB3 locus is linked to increased risk for late onset AD. Presumably, patterns of neuronal and vascular MSRB3 protein expression reflect or underlie pathology associated with AD.	[308]
Astrotactin 2 (ASTN2)	Genetic	-	The ASTN2 gene to play an important role in the developing mammalian brain by forming a complex with its paralog, astrotactin 1 (ASTN1).	The association of ASTN2 genetic variants with age at onset of AD.	[309]
Enoyl-CoA hydratase domain containing 3 (ECHDC3).	Genetic	-	Involved in fatty acid biosynthesis in mitochondria.	GWAS AD with SNPs in *USP6NL/ECHDC3.*	[306]
DNA Methyltransferase 3 Alpha (DNMT3A)	Genetic Epigenetic	-	DNMT3A gene encodes a DNA methyltransferase that is functions in DNA methylation.	An association between the rs1187120 SNP in DNMT3A and the annual decline in cognitive functioning. Presumably, DNMT3A moderates cognitive decline in subjects with MCI.	[187]
Micro RNAs	Epigenetic	Blood CSF	Micro RNAs (miRNAs) are short, non-coding RNAs that regulate gene expression, play an important role in the development of the brain and neurons, and can modulate synaptic plasticity, inflammation, or lipid metabolism.	Altered expression of miRNAs can predict the onset of cognitive dysfunctions. Extracellular circulating miRNAs may reflect early neurodegenerative changes and may predict the onset of MCI/AD at the presymptomatic stage. Serum miR-93 and miR-146a levels are increased in MCI, miR-143 levels are decreased, all these markers are suppressed in AD. MiR-206 and miR-132 upregulate in MCI, and their serum levels also correlate to the degree of cognitive decline. MiR-613 suppresses BDNF expression and levels are elevated in both serum and CSF of AD and MCI patients.	[159,198,260,292,310]
Heme Oxygenase 1 (HMOX1)	Epigenetic	Blood	HMOX1 is a heat shock protein that exists in the endoplasmic reticulum. HMOX1 binds with NADPH cytochrome p450 reductase to convert the pro-oxidant heme to CO, Fe^2+^, and biliverdin.	The methylation status of HMOX1 at a specific promoter CpG site is related to AD progression. The lower methylation of HMOX1 at the -374 promoter CpG site in AD patients compared to MCI and control, and a correlation between neuropsychological score and demethylation level.	[190]
ATP Binding Cassette Subfamily A Member 2 (ABCA2)	Epigenetic	Blood	ABCA2 is transporters are located throughout the brain, with a focus at the BBB, facilitate the strictly regulated influx/efflux of various substances and protect the brain from toxic and harmful compounds.	ABCA2 mRNA expression is upregulated in AD compared with controls. Methylation of 2 of 36 CpG islands in the ABCA2 gene negatively associated with AD risk. ACBA2 mRNA expression could be used to diagnose MCI and Huntington’s disease (HD) and to distinguish HD from AD, but not AD from MCI.	[191]
Receptor for advanced glycation end products (RAGE/AGER)	Epigenetic Inflammatory	-	RAGE is a receptor of the immunoglobulin super family. A critical role of RAGE in AD includes Aβ production and accumulation, the formation of neurofibrillary tangles, failure of synaptic transmission, and neuronal degeneration.	Increased expression of RAGE on the membrane of neurons and microglia is relevant to the pathogenesis of neuronal dysfunction and death of AD (neuroimaging).	[292,311]
Beta-nerve growth factor (β-NGF)	Neuroplasticity	Blood	A neurotrophic factor, that plays a protective role in the development and survival of certain target neurons.	The β-NGF concentration is increased from patients with MCI and mild AD and is decreased in patients with severe AD.	[204,312]
		CSF		The increased levels of NGF in AD patients.	[208]
BDNF	Neuroplasticity	Blood	BDNF is a neurotrophin in the brain.	AD or MCI is accompanied by reduced BDNF levels. The increase in BDNF might reflect a compensatory mechanism against early neurodegeneration and seems to be related to inflammation.	[199,200,203]
DBR	Neuroplasticity	Blood	Dipeptidyl peptidase-4 (DPP4) activity to BDNF ratio	The DBR was positively associated with MCI and may be used as a risk biomarker for MCI in an elderly population with normal glucose tolerance.	[202]
Tropomyosin receptor kinase A (TrKA)	Neuroplasticity	Blood	β-NGF receptor.	The TrKA expression is decreased in monocytes from patients with severe AD and is increased in monocytes from patients with MCI and mild AD.	[204]
p75 neurotrophin receptor (p75NTR)	Neuroplasticity	Blood	β-NGF receptor.	The p75NTR expression is increased in monocytes from patients with severe AD.	[204]
Neurofilament light (NF-L)	Neuroplasticity	CSF Blood	A structural protein present in long axons. The biomarker for axonal degeneration.	The concentration of NF-L is increased in the CSF of AD, especially so in those with rapid disease progression. However, increased NF-L in the CSF is not specific to AD, and is detected in other dementias. Serum and plasma NF-L concentrations correlate with their concentrations in the CSF.	[160,206,276,280,313]
Neurogranin (Ng)	Neuroplasticity	CSF	Ng is a dendritic protein enriched in neurons that is involved in long-term potentiation of synapses, particularly so in the hippocampus and the basal forebrain.	CSF Ng is the best-established biomarker for synapse loss or dysfunction associated with AD. MCI showed a trend towards increased levels of Ng. Elevated Ng levels may predict progression from MCI to AD.	[206,276,292,314,315]
Neuron-specific enolase (NSE)	Neuroplasticity	CSF	NSE is the only currently known common biomarker of all differentiated neurons. The determination of the concentration of NSE in the serum and CSF provides information on the severity of neuronal damage and the integrity of the BBB.	A candidate biomarker for neuronal loss in AD. NSE is elevated in AD.	[276,280,316]
Synaptosomal-associated protein-25 (SNAP-25)	Neuroplasticity	CSF	A marker of functional synapses. Component of the soluble N-ethylmaleimide-sensitive factor attachment protein receptors (SNARE) complex. These proteins mediate synaptic communication by initiating the fusion of synaptic vesicles.	Higher levels SNAP-25 fragments in AD.	[292,317]
Neuronal thread protein (AD7c-NTP)	Neuroplasticity	Urine	A transmembrane phosphoprotein that causes apoptosis and neuritic sprouting in transfected neuronal cells.	Aβ positive subjects showed elevated urine AD7c-NTP level. UrinaryAD7C-NTP in the AD patients is higher than in the non-AD groups.	[272,318,319,320]
TNFα	Inflammatory	Blood	Pro-inflammatory cytokine.	Increased in MCI, that may reflect neuronal dysfunction/loss. The expression of TNF-α can be induced by the amyloid peptide, and its expression can increase during the progression of the disease.	[212,215,292]
Soluble Tumor Necrosis Factor receptors 1 (sTNFR1)	Inflammatory	Blood	The membrane receptor that binds TNFα.	Increased in AD when compared to controls and MCI. Higher levels are associated with a greater risk of progression from normal cognition to MCI.	[203,207,208,209]
Soluble Tumor Necrosis Factor receptors 2 (sTNFr2)	Inflammatory	Blood	The membrane receptor that binds TNFα.	Higher concentrations are associated with a greater global Aβ burden in those with lower hippocampal volume.	[218]
IL-1	Inflammatory	Blood	Cytokine	The IL-1α and IL-1β, their antagonist IL-1Ra, and their soluble receptor sIL-1R1 are increased in AD. The increased levels of IL-1β and IL-1 receptor sIL-1R2 in MCI patients. The lower IL-1β is associated with increasing duration of memory symptoms in the probable-AD patients.	[208,211,214,236]
IL-2	Inflammatory	Blood	Cytokine	Increased in MCI. The lower IL-2 is associated with increasing duration of memory symptoms in the probable-AD group patients.	[214]
IL-4	Inflammatory	Blood	Cytokine	Increased in MCI. The lower IL-4 were associated with increasing duration of memory symptoms in the probable-AD group patients. Plasma IL-4 associate with hippocampal sub-regions volume in MCI ana AD.	[213,214]
IL-6	Inflammatory	Blood	Cytokine	Increased in AD. The higher IL-6 levels is associated with greater odds of an MCI diagnosis. Higher concentrations of IL-6 is associated with greater global Aβ burden in those with lower hippocampal volume.	[208,218,321]
IL-8	Inflammatory	Blood	Chemokine	Decreased in AD. Higher IL-8 is associated with greater cognitive decline in individuals with cognitive deterioration and/or progression to MCI/probably AD.	[208,216]
IL-10	Inflammatory	CSF	Cytokine	Increased in AD.	[208]
		Blood		The increased levels of IL-10 in MCI patients. The higher IL-10 levels are associated with greater odds of an MCI diagnosis. Higher IL-10 is associated with greater cognitive decline in individuals with cognitive deterioration and/or progression to MCI/probably AD.	[214,216,321]
IL-18	Inflammatory	Blood	Cytokine	The IL-18 levels in MCI and AD patients is higher than in the controls, in MCI higher than in AD.	[236]
Alpha1-antichymotrypsin (α1-ACT)	Inflammatory	CSF	The α1-ACT protein is a member of the serpin family of proteins, a group of proteins that inhibit serine proteases.	Increased in AD.	[208]
Complement 3 (C3)	Inflammatory	CSF	Component of the complement system.	A lower level of C3 is associated with faster cognitive decline in MCI.	[228]
Macrophage migration inhibitory factor (MIF)	Inflammatory	CSF	Pro-inflammatory cytokine.	MIF-related inflammation is related to amyloid pathology, tau hyperphosphorylation, and neuronal injury at the early clinical stages of AD. Higher MIF levels are associated with accelerated cognitive decline in MCI and mild dementia.	[230]
S100 Calcium Binding Protein A9 (S100A9)	Inflammatory	CSF	Pro-inflammatory protein	The S100A9 and Aβ(1-42) levels correlate with each other.	[237]
Monocyte chemoattractant protein-1 (MCP-1)	Inflammatory	CSF	Chemokine	Increased in AD patients.	[208]
		Blood		AD patients have higher plasma MCP-1 levels compared with MCI patients and controls, and severe AD patients have the highest levels. A higher plasma MCP-1 level is associated with greater severity and faster cognitive decline. MCP-1 optimally differentiates AD and elderly control, AD, and MCI.	[227,231]
Transforming growth factor-beta (TGF-β)	Inflammatory	CSF	Cytokine	Increased in AD.	[208,322]
		Blood		The level of TGF-β in patients with MCI is increased compared with patients with AD-related dementia.	[233]
Factor H (FH)	Inflammatory	Blood	A major soluble inhibitor of complement.	FH optimally differentiated AD and elderly control.	[227,323]
		CSF		A lower level of FH is associated with faster cognitive decline in MCI.	[228]
Factor B (FB)	Inflammatory	Blood	A component of the alternative pathway of complement activation.	FB optimally differentiated AD and elderly control.	[227]
CRP	Inflammatory	Blood	Acute inflammation protein that is overexpressed in inflammatory conditions.	Increased in AD.	[208,217]
Soluble CD40 ligand (sCD40L)	Inflammatory	Blood	CD40 is a receptor is expressed by B cells, professional antigen-presenting cells, as well as non-immune cells and tumors. CD40 binds its ligand CD40L.	Increased in AD.	[208,324]
Serum amyloid A (SAA)	Inflammatory	Blood	A highly conserved, acute-phase protein synthesized predominantly by the liver.	SAA level is associated with MCI and AD. The increased levels in MCI patients.	[209,238]
Eotaxin-1	Inflammatory	Blood	Chemokine	Eotaxin-1 optimally differentiated AD and elderly control, AD and MCI.	[227]
Myeloperoxidase (MPO)	Inflammatory	Blood	The MPO protein, released from neutrophils azurophilic granules, contributes to formation of cytotoxic ROS.	Concentrations is higher in AD.	[229]
Neutrophil gelatinase associated lipocalin (NGAL)	Inflammatory	Blood	NGAL protein is released on neutrophils degranulation to kill bacterial pathogens.	Concentrations is higher in AD and MCI.	[229]
Macrophage inhibitory cytokine-1, (MIC-1/GDF15)	Inflammatory	Blood	A stress response cytokine and a member of the TGF-β superfamily.	MIC-1/GDF15 levels are associated with cognitive performance and cognitive decline.	[232]
Regulatory T-cells (Tregs)	Inflammatory	Blood	Tregs play an important role in modulating inflammation.	Patients with MCI have stronger Treg-related immunosuppression status compared with patients with probable AD-related dementia.	[233]
Adiponectin	Inflammatory	Blood	Adiponectin is released by the adipose tissue and regulates energy homeostasis and has anti-atherogenic and anti-inflammatory effects.	The levels of adiponectin are lower in MCI and AD as compared to controls. Lower levels are associated with cognitive dysfunction.	[235]
Leukocyte telomere length (LTL)	Inflammatory	Blood	Telomeres are repeated nucleotide sequences located at the ends of each chromosome, that shorten as cells divide.	LTL reduction in MCI and AD patients, with AD patients having a stronger reduction than MCI patients.	[236]
Lactate	Inflammatory	Blood	Plasma lactate is associated with systemic inflammation and may be an indicator of mitochondrial dysfunction.	Higher plasma lactate levels may lead to a higher prevalence of MCI.	[109,292]
Oncostatin M (OSM)	Other	Blood	A member of the IL-6 family cytokines, plays a role in inflammation, autoimmunity, and cancers.	Increased in Aβ+ MCI patients and Aβ- MCI patients compared to Aβ- cognitively normal individuals.	[240,325]
Chitinase 1 (CHIT1)	Other	CSF	Putative marker of microglial activation.	CHIT1 levels are up-regulated in Aβ+ MCI patients, Aβ- MCI patients, and Aβ+ cognitively normal individuals compared to Aβ- cognitively normal individuals.	[240]
SPARC Related Modular Calcium Binding 2 (SMOC2)	Other	CSF	A member of the SPARC protein family, which is involved in microgliosis and functional recovery after cortical ischemia.	SMOC2 levels is increased in Aβ+ MCI patients and Aβ+ cognitively normal individuals compared to Aβ- cognitively normal individuals.	[240]
Matrix Metallopeptidase 10 (MMP-10/Stromelysin-2)	Other	CSF	Proteins of the matrix metalloproteinase family. MMP10 is expressed by macrophages in numerous tissues after injury.	Compared to Aβ- cognitively normal individuals, Aβ+ MCI patients showed increased levels of MMP-1.	[240,326]
Low Density Lipoprotein Receptor (LDLR)	Other	CSF	The LDLR family of proteins is involved in lipoproteins trafficking.	Modestly decreased in Aβ+ MCI patients and Aβ+ cognitively normal individuals.	[240,327]
Eukaryotic Translation Initiation Factor 4E Binding Protein 1 (EIF4EBP1)	Other	CSF	A member of a family of translation repressor proteins. mTOR and Wnt/β-catenin signaling.	Compared to Aβ- cognitively normal individuals, Aβ+ MCI patients showed increased CSF levels of EIF4EBP1.	[240,328]
Roundabout Guidance Receptor 2(ROBO2)	Other	CSF	ROBO2 have been implicated in CNS development, axonal growth, and recovery after nerve injury.	Decreased in the Aβ- MCI patients, compared to controls.	[240]
Repulsive Guidance Molecule BMP Co-Receptor B (RGMB)	Other	CSF	RGMB have been implicated in CNS development, axonal growth, and recovery after nerve injury.	Decreased in the CSF of Aβ- MCI patients, compared to controls.	[240]
Tissue plasminogen activator (tPA)	Other	CSF	A blood-clotting enzyme that co-localizes with amyloid-rich regions and phosphorylated tau in post-mortem AD brains.	Decrease in Aβ+ MCI patients.	[240]
STAM-binding protein (STAMBP)	Other	CSF	Zinc metalloprotease.	Increase in Aβ+ MCI patients.	[240]
Chitinase-3-like protein 1 (YKL-40)	Other	CSF	A chitin-binding lectin which belongs to the glycosyl hydrolase family.	Increased concentrations are observed in the early stages of AD, in fully developed AD, and in MCI. Elevated levels predicted progression from MCI to symptomatic AD and other types of dementia. Interaction in the association between YKL-40 levels and gray-matter volume according to ε4 status. Levels of YKL-40 in MCI predicted progression to AD.	[206,208,221,224,225,226,240,280,292,329]
Synaptotagmin-1	Other	CSF	Pre-synaptic vesicle protein.	Increased in patients with dementia due to AD and in patients with MCI due to AD.	[330]
		Blood		Lower in patients with frontotemporal dementia and AD than in controls. Correlated with cognition assessed.	[331]
OX-2 Membrane Glycoprotein (CD200)	Other	CSF	A glycoprotein previously associated with enhanced in-vivo and in-vitro amyloid phagocytosis.	Modestly decreased in Aβ+ MCI patients and Aβ- MCI patients.	[240]
		Blood		Increased in Aβ+ cognitively normal individuals compared to Aβ- cognitively normal individuals.	
Alpha-2-macroglobulin (α2M)	Other	Blood	Inhibitor a broad spectrum of proteases.	Upregulated in AD.	[243,332]
Apolipoprotein A-1 (ApoA-1)	Other	Blood	The principal protein fraction of high-density lipoprotein (HDL), that plays an important role in lipid transfer.	Involved in Aβ formation. Downregulated in AD.	[243,333]
Metallo-proteinases (MMP-3)	Other	Blood	MMP-3 participates in normal extracellular matrix turnover during embryonic development, organ morphogenesis and wound healing.	Higher in subjective memory impairment, MCI, and probable AD patients than asymptomatic patients.	[216,334]
Vascular endothelial growth factors A (VEGF-A)	Other	Blood	A growth factor with pro-angiogenic activity, having a mitogenic and an anti-apoptotic effect on endothelial cells, increasing the vascular permeability, promoting cell migration.	Decrease levels in MCI.	[208,335]
Plasminogen activator inhibitor 1 (PAI-1)	Other	Blood	A serine protease inhibitor (Serpin), that confers a high risk of vascular diseases. A primary factor in regulating the balance between thrombosis and fibrinolysis.	An effect on reducing brain Aβ clearance. The decreased levels in MCI patients. Levels are higher in cognitively normal and non-amnestic multiple domain MCI than in amnestic multiple domains MCI.	[208,209]
p53	Other	Blood	A transcription factor that both positively and negatively regulates the expression of a large and disparate group of responsive genes.	Misfolded p53 is considered a strong risk factor for MCI progression in AD. Elevated levels of unfolded p53 have been found in both AD and MCI compared to healthy controls. High levels of unfolded p53 in the blood may be a prognostic marker of conversion of MCI to AD.	[292,336,337]
C-peptide	Other	Blood	C-peptide is released together with insulin from the pancreatic beta cells.	The increased levels in MCI patients.	[215,338]
D-amino acid oxidase (DAO)	Other	Blood	A flavoenzyme that degrades D-amino acids, mainly D-serine. Regulatory function on NMDA receptor.	Higher in patients with cognitive decline.	[244]
N-Methyl-D-aspartate glutamate receptor antibodies (NMDAR Abs)	Other	Blood	NMDAR plays a central role in learning and memory and has a potential role in the pathophysiology of neuropsychiatric disorders.	NMDAR Abs are present in different types of dementia and elderly healthy individuals. In combination with disturbed blood-CSF-barrier integrity, they seem to promote their pathological potential on cognitive decline.	[245,246]
Homocysteic acid (HCA)	Other	Blood	HCA is produced from homocysteine by oxidation. HCA exhibits very high brain toxicity at low concentrations and over activated the NMDAR.	HCA is a good candidate of a biomarker of MCI and a good target for treatment of AD.	[247]
Matrix metallopeptidase 9 (MMP-9)	Other	Blood	MMPs are secreted by glial and neuronal cells in the brain and are involved in neuroinflammation and neurotoxicity, which play a role in hippocampal-dependent learning.	Increased in Aβ+ MCI patients and Aβ- MCI patients compared to Aβ- cognitively normal individuals.	[240]
Hydroxyacylglutathione Hydrolase (HAGH)	Other	Blood	Glucose metabolism.	Increased in Aβ+ cognitively normal individuals compared to Aβ- cognitively normal individuals.	[240]
Urokinase-type plasminogen activator (uPA)	Other	Blood	A serine protease involved in tissue remodeling and cell migration.	Decreased in Aβ+ MCI patients, Aβ- MCI patients, and Aβ+ cognitively normal individuals compared to Aβ- cognitively normal individuals.	[240,339]
AXIN1	Other	Blood	The AXIN1 protein function in the canonical Wnt pathway.	Decreased in Aβ+ MCI patients, Aβ- MCI patients, and Aβ+ cognitively normal individuals compared to Aβ- cognitively normal individuals.	[244,340]
Autoantibodies to Aβ and tau protein	Other	Blood	Autoantibodies either exert a protective effect against AD pathology, causing tissue damage through autoimmune reactions, or enhance neuroprotection by inhibiting toxic aggregation and promoting amyloid clearance.	Autoantibodies could be a marker in the differential diagnosis of neurodegenerative diseases that could distinguish people with MCI from those who have developed early AD.	[292,341,342]
Methylglyoxal (MG) and glyoxal (GO)	Other	Blood	MG and GO are the precursors of many advanced glycation end products.	MG and GO levels are higher in MCI. Levels of MG have higher sensitivity to differentiate MCI from controls but not from AD. GO levels differentiate MCI from control and AD groups.	[241]
Eicosapentaenoic acid (EPA)	Other	Blood	Omega-3 polyunsaturated fatty acid.	Lower EPA levels is associated with AD.	[242]
Docosahexaenoic acid (DHA)	Other	Blood	Omega-3 polyunsaturated fatty acid.	Lower DHA levels is associated with AD.	[242]
Reptin	Other	Blood	Reptin is involved in the regulation of gene transcription, remodeling of chromatin, DNA damage sensing and repair, and tumor biology.	Higher reptin levels is associated with AD.	[242]
Low-density lipoprotein (LDL)	Other	Blood	Group of blood lipoprotein. High levels lead to atherosclerosis, which increases the risks of heart attack.	Higher LDL levels is associated with AD.	[242]
Dipeptidyl peptidase-4 (DPP4)	Other	Blood	A serine protease, that expressed on the membranes of many cells like endothelial cells, Stem cells, T and B-lymphocytes and also as soluble form in plasma.	Patients in the highest quartile of DPP4 activity have lower cognitive scores compared with subjects in the lowest quartile. In the highest DPP4 quartile, MCI risk was higher than in the lowest quartile. The risk for MCI increased more with higher levels of DPP4 activity.	[343]
Tryptophan pathway metabolites.	Other	Blood Urine	Tryptophan is the precursor of the monoaminergic neurotransmitter serotonin which exerts both central and peripheral control on numerous physiological functions.	Lower metabolite concentrations of tryptophan pathway metabolites in the AD group: serotonin (urine, serum), 5-hydroxyindoleacetic acid (urine), kynurenine (serum), kynurenic acid (urine), tryptophan (urine, serum), xanthurenic acid (urine, serum), and kynurenine/tryptophan ratio (urine). A decreasing trend in concentrations is observed: control > MCI > AD.	[249]

## Data Availability

Not applicable.

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
