# Peer review of "Neurobiological Highlights of Cognitive Impairment in Psychiatric Disorders"

_ijms, 2022, doi:10.3390/ijms23031217_

Round 1
Reviewer 1 Report
Authors should re-read the whole manuscript as it is subjected to some errors (i.e. unclear sentences like in line 92) and stylistic mistakes (line 92).
Authors should note that the references are not edited according to the journal’s guidelines (should not be written in superscript). Additionally, authors left spaces between a series of references, resulting in a wrong editing (i.e. line 69, it should be [3, 5-8]).
Authors missed out some important concepts on the explored topic. Line 115-117 should be rephrased as the link between chronic conditions underlined by a chronic sub-clinical inflammation and cognitive/affective disorders is bidirectional (please see PMID: 31296155). In fact, cardiovascular conditions may be implicated in the pathogenesis of brain disorders, and not only be a consequence of the latter (please see PMID: 31296155). Another neurobiological mechanism should be underlined in the introduction, namely mitochondrial disfunction, especially as authors included several studies exploring this mechanism in the table (PMID: 31551791).
Finally, since the article focuses on neurobiological aspects of cognitive/affective disorders, authors are advised to include brief note/list on the main neurobiological mechanisms in the introduction for the reader.
Author Response
We would like to thank Reviewer 1 for favorable comments to our work and valuable suggestions to improve the manuscript. We have changed the text of the manuscript considerably in accordance with the comments.
- Authors should re-read the whole manuscript as it is subjected to some errors (i.e. unclear sentences like in line 92) and stylistic mistakes (line 92).
We are grateful to Reviewer 1 for bringing this issue to our attention. We used the MDPI service (https://www.mdpi.com/authors/english) to correct stylistic and linguistic errors.
- Authors should note that the references are not edited according to the journal’s guidelines (should not be written in superscript). Additionally, authors left spaces between a series of references, resulting in a wrong editing (i.e. line 69, it should be [3, 5-8]).
We are grateful to Reviewer 1 for this observation, and we have corrected the references according to the journal's guidelines.
- Authors missed out some important concepts on the explored topic. Line 115-117 should be rephrased as the link between chronic conditions underlined by a chronic sub-clinical inflammation and cognitive/affective disorders is bidirectional (please see PMID: 31296155). In fact, cardiovascular conditions may be implicated in the pathogenesis of brain disorders, and not only be a consequence of the latter (please see PMID: 31296155). Another neurobiological mechanism should be underlined in the introduction, namely mitochondrial disfunction, especially as authors included several studies exploring this mechanism in the table (PMID: 31551791).
We are thankful to Reviewer 1 for this essential question. We have added relevant information to all sections and highlighted in yellow. Page 4-5, 8-9, 12, 19-20.
- Finally, since the article focuses on neurobiological aspects of cognitive/affective disorders, authors are advised to include brief note/list on the main neurobiological mechanisms in the introduction for the reader.
Following a recommendation of Reviewer 1, we have compiled and included in the introduction a list on the main neurobiological mechanisms (Page 4-5).
Reviewer 2 Report
This scientific review aimed to generalize knowledge regarding cognitive impairment, especially among those people with mental health issues. This project contributes to the literature regarding cognitions associated psychiatric disorders. Overall, this paper was written well, but a few things need to be considered and could be improved.
- For introduction, there is a lot of missing reference information. For example, 1) cognitive functioning refers to multiple……, 2) Some cognitive functions may impair as a part of normal aging due to……, 3) Both genetic and environmental factors influence…., etc. Authors need to use the relevant citations to support their content.
- How did authors choose the domains as mentioned above (such as schizophrenia, depression, etc.)? Is there any theorical framework guiding this?
- I think authors need to adjust their logics in writing. For example, 1. Introduction, 2. Schizophrenia, 3. Depression… 2 and 3 do not have parallel relationship with 1. They should use a subtitle for each domain or adjust the titles. Also, it is hard to follow from the beginning to the end, and authors may consider re-writing it by using subtitles and subheads.
- This paper needs a special section regarding the abbreviations of the terminologies.
- In conclusions and future directions, I did not read much for the future directions. So, what are the specific directions are they focusing on?
- Authors may need to consult writing center for their paper with grammars and structures.
Author Response
We thank Reviewer 2 for valuable comments. We added some information according to his recommendations and corrected our manuscript.
- For introduction, there is a lot of missing reference information. For example, 1) cognitive functioning refers to multiple……, 2) Some cognitive functions may impair as a part of normal aging due to……, 3) Both genetic and environmental factors influence…., etc. Authors need to use the relevant citations to support their content.
We are thankful to Reviewer 2 for this observation. We have discussed these issues in more detail below, and have provided reference information [50, 51, 52 and 62] (Page 4, 5).
- How did authors choose the domains as mentioned above (such as schizophrenia, depression, etc.)? Is there any theorical framework guiding this?
We chose these areas of research because our department conducts research activities at N.A. Alekseev Psychiatric Clinical Hospital No. 1 (Moscow, Russia), and in our practice we mostly encounter patients with such diagnoses. Therefore, in our review we decided to cover the issue of cognitive impairment in disorders with which we have had extensive experience.
- I think authors need to adjust their logics in writing. For example, 1. Introduction, 2. Schizophrenia, 3. Depression… 2 and 3 do not have parallel relationship with 1. They should use a subtitle for each domain or adjust the titles. Also, it is hard to follow from the beginning to the end, and authors may consider re-writing it by using subtitles and subheads.
Thanks to Reviewer 2 for bringing this issue to our attention, we have made a correction as recommended. We have brought the texts of each section to a single format, according to the plan:
- Genetic mechanisms, dysfunction of various genes involved in the pathogenesis of mental disorders;
- Epigenetic mechanisms - dysfunction of DNA methylation, histone acetylation, and micro RNA (miRNA) content regulating gene expression;
- Dysfunction of neurotransmitter systems;
- Impaired neuroplasticity and synthesis of neurotrophic factors that support nerve cell function;
- Neuroinflammation;
- Vascular pathology;
- Mitochondrial dysfunction.
However, we did not label these subsections with appropriate subheadings; in the text, the subheadings begin with a new paragraph.
- This paper needs a special section regarding the abbreviations of the terminologies.
Following a recommendation of Reviewer 2, we have included abbreviations of the terminologies at the end of the manuscript (Page 44-45).
- In conclusions and future directions, I did not read much for the future directions. So, what are the specific directions are they focusing on?
We thank the Reviewer 2 for his valuable comments. We have restated the conclusion and added more information about future research.
- Authors may need to consult writing center for their paper with grammars and structures.
We are grateful to Reviewer 2 for bringing this issue to our attention. We used the MDPI service (https://www.mdpi.com/authors/english) to correct stylistic and linguistic errors.
Round 2
Reviewer 2 Report
I was a reviewer on the original paper. Thanks to the authors for studiously modifying the paper. I believe it is much improved. While I still have a few suggestions before this paper is accepted.
A. For “Neurobiological mechanisms leading to cognitive impairment in various neuro- 155 psychiatric diseases include:”, authors could merge them into the main content in the following paragraphs. This reads repeatedly.
- Genetic mechanisms, dysfunction of various genes involved in the pathogenesis of mental disorders
- Epigenetic mechanisms - dysfunction of DNA methylation, histone acetylation, and micro RNA (miRNA) content regulating gene expression
- Dysfunction of neurotransmitter systems
- Impaired neuroplasticity and synthesis of neurotrophic factors that support nerve cell function.
- Neuroinflammation
- Vascular pathology
- Mitochondrial dysfunction.
B. I suggest using other subtitle marks/numbers such as Roma numbers. There are too many 1, 2, 3, 4, 5….
Author Response
We thank Reviewer 2 for his positive review of our paper. We have corrected the list of neurobiological mechanisms as recommended (Pages 4-5).